# Efficient Recurrent Off-Policy RL Requires a Context-Encoder-Specific Learning Rate

**Fan-Ming Luo**[1,2]    **Zuolin Tu**[2]    **Zefang Huang**[1]    **Yang Yu**[1,2*]

[1] National Key Laboratory for Novel Software Technology, Nanjing University, China
School of Artificial Intelligence, Nanjing University, China
[2] Polixir.ai
luofm@lamda.nju.edu.cn, zuolin.tu@polixir.ai, zf.frank.huang@gmail.com,
yuy@nju.edu.cn

## Abstract

Real-world decision-making tasks are usually partially observable Markov decision processes (POMDPs), where the state is not fully observable. Recent progress has demonstrated that recurrent reinforcement learning (RL), which consists of a context encoder based on recurrent neural networks (RNNs) for unobservable state prediction and a multilayer perceptron (MLP) policy for decision making, can mitigate partial observability and serve as a robust baseline for POMDP tasks. However, prior recurrent RL algorithms have faced issues with training instability. In this paper, we find that this instability stems from the autoregressive nature of RNNs, which causes even small changes in RNN parameters to produce large output variations over long trajectories. Therefore, we propose **R**ecurrent Off-policy RL with Context-**E**ncoder-**Sp**ecific **L**earning Rate (RESeL) to tackle this issue. Specifically, RESeL uses a lower learning rate for context encoder than other MLP layers to ensure the stability of the former while maintaining the training efficiency of the latter. We integrate this technique into existing off-policy RL methods, resulting in the RESeL algorithm. We evaluated RESeL in 18 POMDP tasks, including classic, meta-RL, and credit assignment scenarios, as well as five MDP locomotion tasks. The experiments demonstrate significant improvements in training stability with RESeL. Comparative results show that RESeL achieves notable performance improvements over previous recurrent RL baselines in POMDP tasks, and is competitive with or even surpasses state-of-the-art methods in MDP tasks. Further ablation studies highlight the necessity of applying a distinct learning rate for the context encoder. Code is available at `https://github.com/FanmingL/Recurrent-Offpolicy-RL`.

## 1 Introduction

In many real-world reinforcement learning (RL) tasks [1], complete state observations are often unavailable due to limitations such as sensor constraints, cost considerations, or task-specific requirements. For example, visibility can be obstructed by obstacles in autonomous driving [2, 3] or robotic manipulation [4], and measuring ground friction may be infeasible when controlling quadruped robots on complex terrain [5]. These scenarios are common and typically conceptualized as Partially Observable Markov Decision Processes (POMDPs). Traditional RL struggles with POMDPs due to the lack of essential state information [6].

A mainstream class of POMDP RL algorithms infers unobservable states by leveraging historical observation information, either explicitly or implicitly [6]. This often requires memory-augmented

---

*Yang Yu is the corresponding author.

network architectures, such as recurrent neural networks (RNNs) [7, 8] and Transformers [9]. Replacing standard networks in RL algorithms with these memory-based structures has proven effective in solving POMDP problems [10, 11, 12, 13]. Particularly, recurrent RL [11], which employs an RNN-based context encoder to extract unobservable hidden states and an MLP policy to make decisions based on both current observation and hidden states, demonstrates robust performance in POMDP tasks. Compared to Transformer-based RL [13], recurrent RL offers lower inference time complexity, making it highly applicable, especially in resource-constrained terminal controllers.

Despite these advantages, recurrent RL faces a significant challenge: while advanced RNN architectures [7, 8, 14] can effectively address gradient explosion issues, training in recurrent RL often remains more unstable compared to MLP-based RL, particularly when handling long sequence lengths. This instability can lead to poor policy performance and even training divergence [15]. Although existing methods usually avoid training with full-length trajectories by using shorter trajectory segment [11], this practice introduces distribution shift due to the inconsistency between the sequence lengths used during training and deployment. ESCP [16] addressed this by truncating the history length during deployment to match the training sequence length, but this may limit policy performance due to restricted memory length. Additionally, some methods introduce auxiliary losses to aid context encoders in learning specific information. For instance, they train alongside a transition model, forcing the outputs of context encoder to help minimize the prediction error of this model [17, 18, 19]. However, these methods require the RNN to learn specific information, which may limit the RNN's potential and may only be applicable to a subset of tasks.

In this work, we found that, with the autoregressive property of RNNs, the output variations caused by parameter changes are amplified as the sequence length increases. As RNN parameters change, even slight variations in the RNN output and hidden state at the initial step can become magnified in subsequent steps. This occurs because the altered hidden state is fed back into the RNN at each step, causing cumulative output variations. Our theoretical analysis shows that these output variations grow with the sequence length and eventually converge. The amplified output variations can lead to instability in the RL process. For instance, in off-policy RL algorithms, the bootstrapped update target of the Q-function may fluctuate significantly, resulting in unstable Q-function training. To avoid training instability caused by excessive amplification of output variations, we propose **R**ecurrent Off-policy RL with Context-**E**ncoder-**Sp**ecific **L**earning Rate (RESeL). Specifically, we employ a lower learning rate for the RNN context encoder while keeping the learning rate for the other MLP layers unchanged. This strategy ensures efficient training for MLPs, which do not experience the issue of amplified output variations.

Based on the SAC framework [20], and incorporating context-encoder-specific learning rate as well as the ensemble-Q mechanism from REDQ [21] for training stabilization, we developed the practical RESeL algorithm. We empirically evaluated RESeL across 18 POMDP tasks, consisting of classic POMDP, meta-RL, credit assignment scenarios, as well as five MDP locomotion tasks. In our experiments, we first observed the increasing RNN output variations over time and demonstrated that the context-encoder-specific learning rate can mitigate the amplification issue, significantly enhancing the stability and efficacy of RL training. Comparative results indicate that RESeL achieves notable performance improvements over previous recurrent RL methods in POMDP tasks and is competitive with, or even surpasses, state-of-the-art (SOTA) RL methods in MDP tasks. Further ablation studies highlight the necessity of applying a distinct learning rate for the context encoder.

## 2 Background

**Partially Observable Markov Decision Processes** (POMDPs) [22, 6] enhance MDPs for scenarios with limited state visibility, addressing a number of real-world decision-making challenges with imperfect or uncertain data. A POMDP is defined by $\langle \mathcal{S}, \mathcal{A}, \mathcal{O}, P, O, r, \gamma, \rho_0 \rangle$, consisting of state space $\mathcal{S}$, action space $\mathcal{A}$, observation space $\mathcal{O}$, state transition function $P$, observation function $O$, reward function $r$, discount factor $\gamma$, and initial state distribution $\rho_0$. Unlike MDPs, agents in POMDPs receive

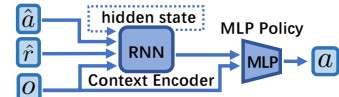

Figure 1: A simple recurrent policy architecture.

observations $o \in \mathcal{O}$ instead of direct state information, requiring a belief state—a probability distribution over $\mathcal{S}$—to inform decisions. The objective is to search for a policy $\pi$ to maximize the expected rewards, while factoring in observation and transition uncertainties.

**Recurrent RL** [23, 11] involves employing RNN-based policy or critic models within the framework of RL. Figure 1 provides a commonly utilized recurrent policy architecture. This architecture has two main components. First, a context encoder processes the current observation $o$, the last action $\hat{a}$, the reward $\hat{r}^2$, and the RNN hidden state. Then, an MLP policy uses the context embedding and $o$ to generate actions, facilitating the extraction of non-observable states into a context embedding. We can train the recurrent policy by incorporating it into any existing RL algorithm. During training, recurrent RL requires batch sampling and loss computation on a trajectory-by-trajectory basis.

## 3 Related Work

In this work, we focus on recurrent RL methods [24, 25, 23], which leverage RNN-based models for RL tasks. These methods have demonstrated robust performance in POMDP [10, 22] and meta-RL tasks [25, 26], having shown notable success in practical applications such as robotic motion control [27, 28, 29] and MOBA games [30].

Recurrent RL utilizes RNNs such as GRUs [8], LSTMs [7], and more recently, SSMs [12] to construct policy and critic models. Typically, in recurrent actor-critic RL, both policy and critic adopt RNN architectures [23, 11]. When full state information is available during training, an MLP critic can also be used instead [29], receiving the full state as its input. In this work, we do not assume that complete state information can be accessed, so policy and critic models are both RNN structures.

The most direct way to train RNN models in recurrent RL is combining an RNN structure with existing RL algorithms, such as being integrated into on-policy methods like PPO [28, 30, 12, 31] and TRPO [25, 32] as well as off-policy methods like TD3 [11] and SAC [23]. This work follows the line of recurrent off-policy RL [11, 23], as we believe that existing recurrent off-policy RL algorithms have not yet achieved their maximum potential.

Despite the simplicity of direct integration, however, training RNNs often suffers from the training instability issue [12, 15], especially in long sequence length. Previous methods usually use truncated trajectory to train RNNs [11]. However, deploying recurrent models with full-trajectory lengths can result in a mismatch between training and deployment. ESCP [16] addressed this issue by using the same sequence lengths in both training and deployment by truncating the historical length in deployment scenarios, but this could impair policy performance due to the restricted memory length. On the other hand, many studies also employed auxiliary losses in addition to the RL loss for RNN training [19]. These losses aim to enable RNNs to stably extract some certain unobservable state either implicitly or explicitly. For instance, they might train the RNNs to predict transition parameters [33], distinguish between different tasks [16, 34], or accurately predict state transitions [17, 18, 19]. These methods require the RNN to learn specific information, which may limit the RNN's potential and may only be applicable to a subset of tasks, reducing the algorithmic generalizability.

RESeL directly trains recurrent models using off-policy RL due to its simplicity, flexibility, and potential high sample efficiency. The most relevant study to RESeL is [11], which focused on improving recurrent off-policy RL by analyzing the hyperparameters, network design, and algorithm choices. More in depth, RESeL studies the instability nature of previous methods and enhances the stability of recurrent off-policy RL through a specifically designed learning rate.

## 4 Method

In this section, we address the training stability challenges of recurrent RL. We will first elaborate on our model architecture in Sec. 4.1 and introduce how we address the stability issue in Sec. 4.2. Finally, we will summarize the training procedure of RESeL in Sec. 4.3.

### 4.1 Model Architectures

The architectures of RESeL policy and critic models are depicted in Fig. 2. Specifically, RESeL policy initially employs MLPs as pre-encoders to map the current and last-step observations ($o$ and $\hat{o}$ respectively), actions ($\hat{a}$), and rewards ($\hat{r}$) into hidden spaces with equal dimensions, ensuring a balanced representation of various information types. These pre-encoded inputs are then concatenated

---

²The inclusion of reward ($\hat{r}$) in policy inputs is optional and depends on the RL task throughout the paper.

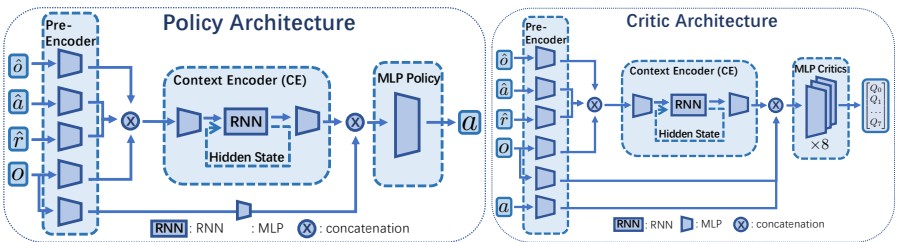

Figure 2: Policy and critic architectures of RESeL.

into a vector and inputted into a context encoder. For the context encoder, RESeL utilizes RNN such as GRU [8] and Mamba [14], to mix sequential information and extract non-observable state information from the observable context. The MLP policy comprises a two-layer architecture, each layer consisting of 256 neurons. The architecture of the pre-encoder and context encoder for the RESeL critic mirrors that of the policy. To improve training stability, RESeL adopts the ensemble-Q technique from REDQ [21], employing 8 MLP critic models, and each one of them comprises two hidden layers with 256 neurons.

## 4.2 Stabilizing Training with a Context-Encoder-Specific Learning Rate

To locate the source of instability, we first formalize the RNN and its sequence generation process. An RNN network $f^\theta(x_t, h_t) : \mathcal{X} \times \mathcal{H} \to \mathcal{Y} \times \mathcal{H}$ is defined as: $y_t, h_{t+1} = f^\theta(x_t, h_t)$, where $\theta$ represents the network parameters, $x_t$, $y_t$, and $h_t$ are the RNN input, output, and hidden state, respectively. For simplicity, we denote $y_t = f_y^\theta(x_t, h_t)$ and $h_{t+1} = f_h^\theta(x_t, h_t)$. Given an input sequence $\{x_0, x_1, \ldots, x_T\}$ of length $T + 1$ and an initial hidden state $h_0$, the output sequence generated by $f^\theta$ is denoted as $\{y_0, y_1, \ldots, y_T\}$. Let $\theta'$ be the parameter neighboring $\theta$ after a one-step gradient update. The output sequence produced by $f^{\theta'}$ is denoted as $\{y'_0, y'_1, \ldots, y'_T\}$. With certain assumptions, we can derive Proposition 1, which bounds the difference between $y_t$ and $y'_t$ at any time step $t$.

**Proposition 1.** *Assuming $f^\theta$ and $f^{\theta'}$ both satisfy Lipschitz continuity, i.e., for all $\hat\theta \in \{\theta, \theta'\}$, $x \in \mathcal{X}$, $h, h' \in \mathcal{H}$, there exist constants $K_h \in [0, 1)$ and $K_y \in \mathbb{R}$ such that:*

$$\left\| f_h^{\hat\theta}(x, h) - f_h^{\hat\theta}(x, h') \right\| \le K_h \left\| h - h' \right\|, \quad \left\| f_y^{\hat\theta}(x, h) - f_y^{\hat\theta}(x, h') \right\| \le K_y \left\| h - h' \right\|,$$

*and for all $x \in \mathcal{X}$, $h \in \mathcal{H}$, the output differences between the RNN parameterized by $\theta$ and $\theta'$ are bounded by a constant $\epsilon \in \mathbb{R}$:*

$$\max \left( \left\| f_h^\theta(x, h) - f_h^{\theta'}(x, h) \right\|, \left\| f_y^\theta(x, h) - f_y^{\theta'}(x, h) \right\| \right) \le \epsilon,$$

*the RNN output difference at $t$-th step is bounded by*

$$\| y_t - y'_t \| \le \underbrace{K_y \frac{1 - K_h^t}{1 - K_h}}_{\text{amplification factor}} \epsilon + \epsilon, \quad \forall t \ge 0. \tag{1}$$

Refer to Appendix A for a detailed proof. Proposition 1 focuses on the case where $K_h \in [0, 1)$, as most RNN models adhere to this condition. GRU [8] and LSTM [7] meet this requirement through the use of sigmoid activation functions, while Mamba [14] achieves it by constraining the values in the hidden state's transition matrix.

In Eq. (1), $\epsilon$ represents the maximum variation of the model output when the model parameters are modified and both the model input and hidden state remain constant, i.e., the first-step output variation. However, from Eq. (1), we observe that due to the variability in the hidden state, the final variation in the model output is amplified. This amplification factor is minimal at $t = 0$, being 0. As $t$ increases, this amplification factor gradually increases and ultimately converges to $\beta := \frac{K_y}{1 - K_h}$. Additionally, it can be verified that as $t$ increases, the upper bound of the average variation in network output $\frac{1}{t} \sum_{i=0}^{t-1} \| y_i - y'_i \|$ also increases with time step $t$ and eventually converges to $\beta\epsilon + \epsilon$ (proved

in Appendix B), which is again amplified by $\beta$. This conclusion indicates that, compared to the case of a sequence length of $1$, in scenarios with longer sequence lengths, the average variations in the RNN output induced by gradient descent are amplified.

This issue is different from the gradient explosion that commonly occurs in RNNs over long sequences [7]. Proposition 1 indicates that the variation in network output is amplified by a constant factor, rather than undergoing exponential explosion. This type of instability is not very significant in supervised learning but can lead to instability in RL processes. For instance, in off-policy RL algorithms, the optimization target of the Q-function minimizes the difference with a bootstrapped target. If the network output varies greatly, this target will fluctuate dramatically, making the Q-function training unstable, which in turn can lead to overall training instability.

To ensure stable training, it is required to use a smaller learning rate to counterbalance the amplified variations in network output caused by long sequence lengths, as a smaller learning rate can help reduce $\epsilon$. However, the output variations in MLPs are not amplified in the same way as in RNNs. If MLPs are also set with a small learning rate, their learning efficiency could be significantly reduced, as the learning rate may become too slow for MLPs, resulting in inefficient training and poor overall performance. Consequently, we propose to use a context-encoder-specific learning rate for the context encoder. In our implementation, we use a smaller learning rate $\text{LR}_{\text{CE}}$ for the context encoder, which contains the RNN architecture. For other layers, we use a normal learning rate $\text{LR}_{\text{other}}$, e.g. $3 \times 10^{-4}$. Applying different learning rates to different modules is similar to Two-Timescale Network [35], which has been shown to improve the training convergence.

### 4.3 Training Procedure of RESeL

Based on the SAC framework [20], combining context-encoder-specific learning rate and the ensemble-Q mechanism from REDQ [21], we developed the practical RESeL algorithm. Specifically, we initially configure context-encoder-specific optimizers for both the policy and value models. After each interaction with the environment, RESeL samples a batch of data from the replay buffer, containing several full-length trajectories. This batch is used to compute the critic loss according to REDQ, and the critic network is optimized using its context-encoder-specific optimizer. Notably, unlike REDQ, we did not adopt a update-to-data ratio greater than 1. Instead of that, we update the network model once per interaction. The policy network is updated every two critic updates. During the policy update, the policy loss from REDQ is used, and the policy network is optimized with its context-encoder-specific optimizer. Here, we delay the policy update by one step to allow more thoroughly training of the critic before updating the policy, which is inspired by TD3 [36] to improve the training stability. The detailed algorithmic procedure is provided in Appendix C and Algorithm 1.

## 5 Experiments

To validate our argument and compare RESeL with other algorithms, we conducted a series of validation and comparison experiments across various POMDP environments. We primarily considered three types of environments: classical partially observable environments, meta-RL environments, and credit assignment environments. To assess the generalizablity of RESeL to MDP tasks, we also test RESeL in five MuJoCo [37] locomotion tasks with full observation provided. We implement the policy and critic models with a parallelizable RNN, i.e., Mamba [14] to accelerate the training process. A detailed introduction of the network architecture can be found in Appendix D.3.2.

In all experiments, the RESeL algorithm was repeated six times using different random seeds, i.e. 1–6. All experiments were conducted on a workstation equipped with an Intel Xeon Gold 5218R CPU, four NVIDIA RTX 4090 GPUs, and 250GB of RAM, running Ubuntu 20.04. For more detailed experimental settings, please refer to Appendix D.

### 5.1 Training Stability

**The updates of the RNN lead to large variations in the model output.** In Sec. 4.2, we observe that the variations in model output induced by a model update are amplified. In this part, we quantify the amplification factor, assess its impact on RL training, and discover how a context-encoder-specific learning rate addresses this issue. We use a POMDP task, specifically WalkerBLT-V-v0, to validate our argument. We loaded a policy model trained by RESeL and updated it with various $\text{LR}_{\text{CE}}$ and

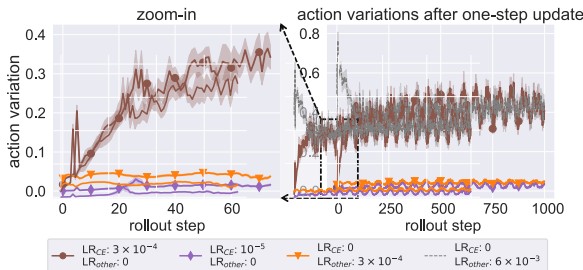

Figure 3: Action variations as the rollout step increases after a single gradient update with different values of $LR_{CE}$ and $LR_{other}$. Action variation refers to the change in policy output after the gradient update compared to its output before the update, using the same input sequences.

$LR_{other}$ for a one-step gradient update. The output differences for all rollout steps between the pre- and post-update models are presented in Fig. 3. The right panel shows the variations in model output as the rollout step increases, while the left panel zooms in on the rollout steps from 0 to 75.

In the zoom-in panel, comparing the brown-circle and yellow-triangle curves reveals that at the first rollout step, the two curves are very close. This indicates that with the same learning rate, identical input, and hidden states, the updates of RNN and MLP have almost the same impact on the model's output. However, as the rollout steps increase, the brown-circle curve gradually rises, increasing by 400% at the 20-th step. This observation aligns with Proposition 1, demonstrating that changes in the hidden state lead to increasing variations in the model's output. With further increases in rollout length, the right panel shows that the brown-circle curve converges at around 0.4 after 250 steps, indicating that the amplification of the output variations eventually converges. This also meets Proposition 1. Ultimately, the increase in action variations is approximately tenfold.

This action variation magnitude is equivalent to an MLP policy network (the gray-dashed curve) with a learning rate increased twentyfold. At this point, the learning rate reaches 0.006, which is typically avoided in RL scenarios due to the risk of training instability. Conversely, reducing $LR_{CE}$ to $10^{-5}$ (the purple-diamond curve) significantly suppresses the amplification of variations. The right panel shows that the orange and purple curves remain at similar levels until the final time step.

**Large output variations lead to instability in RL training.** To investigate how the large variations influence the RL process, we visualized the gradient norm and value function loss during training for two POMDP tasks, as shown in Figs. 4 and 5. We fixed $LR_{other} = 3 \times 10^{-4}$ and compared the training processes of $LR_{CE} = 10^{-5}$ (red line) and $3 \times 10^{-4}$ (orange line). For the latter, we applied gradient clipping; otherwise, training would diverge and stop early. The orange line in Fig. 4 shows the norm of the policy gradient before gradient rescaling, where significant oscillations appear in the gradient in the later stages of training. The orange gradient norm is eventually scaled to 0.5, consistently smaller than the red line, which does not show oscillations. This suggests that the late-stage instability of the orange line does not stem from large gradient magnitudes.

On the other hand, as seen in Fig. 5, the value loss for the orange line remains consistently high, with extremely large values appearing in later stages. This is due to large variations in the RNN output, which lead to an unstable bootstrapped update target for the value function. Consequently, the value loss diverges, causing the abnormal gradient norm observed in Fig. 4. The results in Figs. 4 and 5 reveal the unique challenges of using RNN structures in RL, where traditional RNN stabilization techniques may no longer be effective.

**Using a context-encoder-specific learning rate improves training stability, while traditional RNN stabilization techniques fall short.** To further examine the impact of a lower learning rate on the stability of RL training, we conducted experiments across several POMDP tasks. Building on the setup in Fig. 4, we further evaluated a variant without the gradient clipping technique and another that replaces gradient clipping with a truncation of recurrent backpropagation steps to 32 (green line). For reference, we also included a variant where the CE output is entirely masked to zero (purple line).

Comparing the vanilla RNN-based RL (blue line) with the purple line shows that introducing an RNN can partially address partial observability issues and improve policy performance. However, the blue line still exhibits instability, even triggering early stopping due to outliers. Setting a specific

learning rate for the CE (red line) significantly enhances training stability, indicating that reducing the RNN learning rate indeed improves overall RL training stability. However, we also found that traditional RNN stabilization techniques do not reliably improve RL training stability. For example, truncating the number of recurrent backpropagation steps (green line) can still lead to early stopping, while clipping the gradient norm (orange line) prevents early stopping but does not improve the policy performance. This is because these two stabilization techniques can only suppress gradient explosions in the RNN, keeping the RNN gradients within a normal range. However, the output of the RNN can still vary significantly, which continues to cause instability in RL.

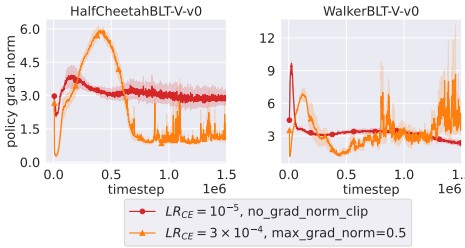

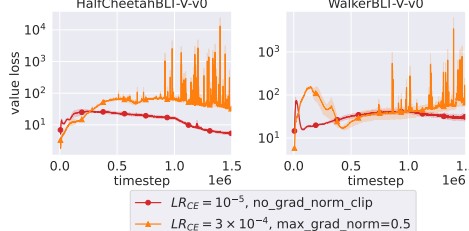

Figure 4: L2-norm of the policy gradient for different RNN stabilization approaches.

Figure 5: Value losses in log-scale for different RNN stabilization approaches.

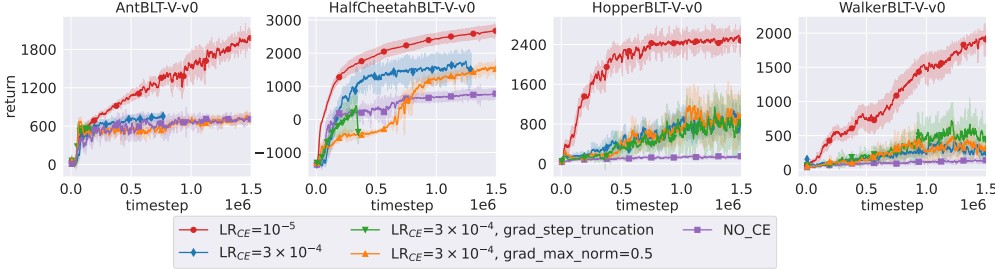

Figure 6: Learning curves in four POMDP tasks with different learning rates and RNN stabilization techniques, shaded with one standard error. We fixed $LR_{other} = 3 \times 10^{-4}$. Some learning curves in AntBLT-V and HalfCheetahBLT-V are incomplete as some runs encountered infinite or NaN outputs.

## 5.2 Performance Comparisons

In this part, we compare RESeL with previous methods in various tasks. The detailed introduction of the baselines and tasks can be found in Appendix D.2 and Appendix D.1. More comparative results can be found in Appendix E.3.

**Classic POMDP Tasks.** We first compare RESeL with previous baselines in four PyBullet locomotion environments: AntBLT, HalfCheetahBLT, HopperBLT, and WalkerBLT. To create partially observable tasks, we obscure part of the state information as done in previous studies [18, 11, 15], preserving only position (-P tasks) or velocity (-V tasks) information from the original robot observations. We benchmark RESeL against prior model-free recurrent RL (MF-RNN) [11], VRM [18], and GPIDE [38]. GPIDE, which extracts historical features inspired by the principle of PID [39] controller, is the state-of-the-art (SOTA) method for these tasks.

The comparative results are shown in Fig. 7. RESeL demonstrates significant improvements over previous recurrent RL methods (MF-RNN, PPO-GRU, and A2C-GRU) in almost all tasks, except for HalfCheetahBLT-P where its performance is close to that of MF-RNN. These results highlight the advantage of RESeL in classic POMDP tasks over previous recurrent RL methods. Furthermore, RESeL outperforms GPIDE in most tasks, establishing it as the new SOTA method. The learning curves of RESeL are more stable than that of GPIDE, suggesting that fine-grained feature design could also introduce training instability, while a stably trained RNN can even achieve superior performance.

**Dynamics-Randomized Tasks.** Instead of directly obscuring part of the immediate state, meta-RL considers a different type of POMDP. In meta-RL scenarios [26], the agent learns across various

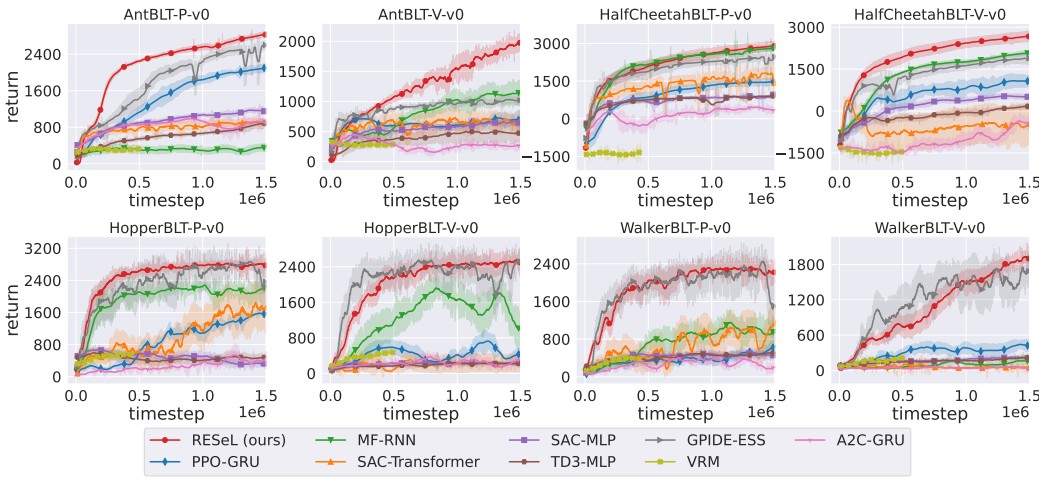

Figure 7: Learning curves shaded with one standard error in classic POMDP tasks.

tasks with different dynamics or reward functions [16, 19], where the parameters of the dynamics and reward functions are not observable. We first compare RESeL with previous methods in the dynamics randomization meta-RL tasks. Following previous meta-RL work [16], we randomized the gravity in MuJoCo environments [37]. We created 60 dynamics functions with different gravities, using the first 40 for training and the remaining for testing. The gravity is unobservable, requiring the agents to infer it from historical experience. We compare RESeL to various meta-RL methods in these tasks, including ProMP [40], ESCP [16], OSI [33], EPI [41], and PEARL [42]. Notably, ESCP, EPI, and OSI use recurrent policies with different auxiliary losses for RNN training. ESCP and PEARL are previous SOTA methods for these tasks. The comparative results are shown in Fig. 8.

We find that in Ant, Humanoid, and Walker2d, RESeL demonstrates significant improvement over all other methods. In Hopper, RESeL performs on par with ESCP, while surpassing other methods. Specifically, RESeL outperforms SAC-RNN by a large margin, further highlighting its advantages. Moreover, ESCP, EPI, and OSI use RNNs to extract dynamics-related information, with ESCP having inferred embeddings highly related to the true environmental gravity. The advantages of RESeL over these methods suggest that the context encoder of RESeL may extract not only gravity (see Appendix E.1) but also other factors that help the agent achieve higher returns.

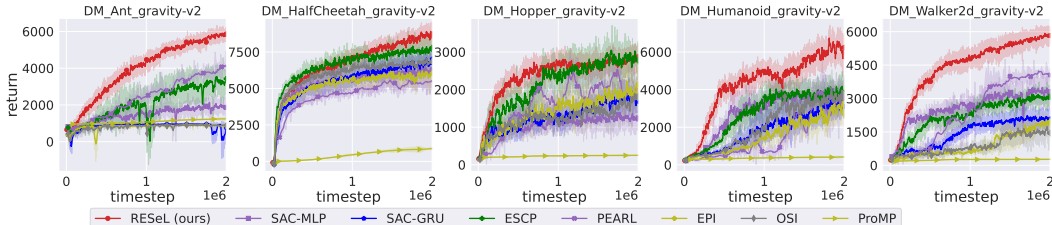

Figure 8: Learning curves shaded with one standard error in dynamics-randomized tasks.

**Classic meta-RL Tasks.** We further compare RESeL with baselines in other meta-RL tasks, particularly the tasks possessing varying reward functions, sourced from [19, 11]. In AntDir, CheetahDir, and HalfCheetahVel, the robots' target moving direction or target velocity are changeable and unobservable. Agents must infer the desired moving direction or target velocity from their historical observations and rewards. Wind is a non-locomotion meta-RL task with altered dynamics. We compare RESeL to previous meta-RL methods, including RL$^2$ [25], VariBad [19], and MF-RNN [11].

The learning curves of these methods are shown in Fig. 9. In the AntDir and HalfCheetahDir tasks, we found that RESeL can converge within 5M steps. Our results show that RESeL demonstrates high sample efficiency and superior asymptotic performance in AntDir, CheetahDir, and HalfCheetahVel. In Wind, RESeL performs comparably to MF-RNN, as this task is less complex. These findings suggest that RESeL effectively generalizes to POMDP tasks with hidden reward functions.

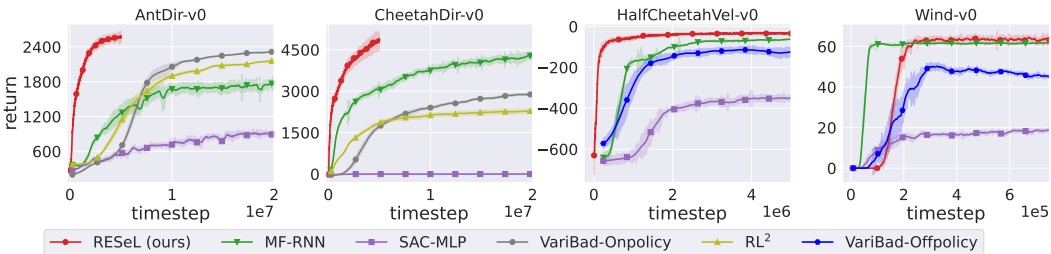

Figure 9: Learning curves shaded with one standard error in meta-RL tasks.

**Credit Assignment Tasks.** A notable application of recurrent RL is solving credit assignment problem [11, 15]. We also compare RESeL with MF-GPT [15] and MF-RNN [11] in a credit assignment task, namely Key-to-Door. In this task, it is required to assign a reward obtained at the last-step to an early action. We compared RESeL with algorithms MF-GPT and MF-RNN. As did in [15], we test RESeL with credit assignment length of $[60, 120, 250, 500]$. the hardness of the task grows as the length increases. In this task, the methods are evaluated by success rate. The results are shown in Fig. 10. The success rate of RESeL matches the previous SOTA MF-GPT in tasks with [60,120] lengths, closed to 100% success

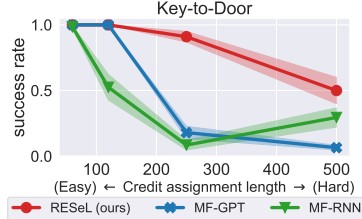

Figure 10: Success rate in the key-to-door task with different credit assignment lengths.

rate. For harder tasks with the lengths being [250, 500], RESeL reached higher success rates than the others. The results indicate that RESeL outperforms Transformer-based methods in handling challenging credit assignment tasks.

**Classic MuJoCo Locomotion Tasks.** Finally, we would like to discover how RESeL performs in MDP tasks. We adopt five classic MuJoCo tasks and compare RESeL with previous RL methods, e.g., SAC [20], TD3 [36], and TD7 [43]. As the current SOTA method for these environment, TD7 introduces several enhancements based on TD3, e.g., a representation learning method and a checkpoint trick, which are not existing in RESeL. The results, listed in Table 1, show that RESeL is comparable to TD7, showing only a 4.8% average performance drop. This demonstrates that RESeL is effective enough to nearly match the performance of the most advanced MDP algorithm. In Hopper and Walker, RESeL even surpasses TD7 by a significant margin. By comparing RESeL with SAC, we find RESeL is superior to SAC in all tasks and can improve SAC by $34.2\%$ in average. These results indicate that RESeL can also be effectively extended to MDP tasks with notable improvement.

Table 1: Average performance on the classic MuJoCo tasks at 5M time steps $\pm$ standard error.

|  | TD3 | SAC | TQC | TD3+OFE | TD7 | RESeL (ours) |
|---|---|---|---|---|---|---|
| HalfCheetah-v2 | $14337 \pm 1491$ | $15526 \pm 697$ | $17459 \pm 258$ | $16596 \pm 164$ | $\mathbf{18165 \pm 255}$ | $16750 \pm 432$ |
| Hopper-v2 | $3682 \pm 83$ | $3167 \pm 485$ | $3462 \pm 818$ | $3423 \pm 584$ | $4075 \pm 225$ | $\mathbf{4408 \pm 5}$ |
| Walker2d-v2 | $5078 \pm 343$ | $5681 \pm 329$ | $6137 \pm 1194$ | $6379 \pm 332$ | $7397 \pm 454$ | $\mathbf{8004 \pm 150}$ |
| Ant-v2 | $5589 \pm 758$ | $4615 \pm 2022$ | $6329 \pm 1510$ | $8547 \pm 84$ | $\mathbf{10133 \pm 966}$ | $8006 \pm 63$ |
| Humanoid-v2 | $5433 \pm 245$ | $6555 \pm 279$ | $8361 \pm 1364$ | $8951 \pm 246$ | $\mathbf{10281 \pm 588}$ | $10490 \pm 381$ |
| Average | 6824 | 7109 | 8350 | 8779 | **10010** | 9532 |

### 5.3 Sensitivity and Ablation Studies

**Sensitivity Studies.** In this section, we analyze the impact of $\mathrm{LR_{CE}}$ and $\mathrm{LR_{other}}$ with experiments on the WalkerBLT-V task. Initially, we fixed $\mathrm{LR_{other}} = 3 \times 10^{-4}$ and varied $\mathrm{LR_{CE}}$. The final returns for different $\mathrm{LR_{CE}}$ values are shown in Fig. 11a. The figure demonstrates that the highest model performance is achieved when $\mathrm{LR_{CE}}$ is between $5 \times 10^{-6}$ to $10^{-5}$. Both larger and smaller $\mathrm{LR_{CE}}$ values result in decreased performance. Notably, an overly small $\mathrm{LR_{CE}}$ is preferable to an overly large one, as a small learning rate can slow down learning efficiency, whereas a large learning rate can destabilize training, leading to negative outcomes. Next, we fixed $\mathrm{LR_{CE}} = 10^{-5}$ and varied $\mathrm{LR_{other}}$. The resulting final returns are presented in Fig. 11b. Here, the model achieves the highest score with $\mathrm{LR_{other}} = 3 \times 10^{-4}$, while other values for $\mathrm{LR_{other}}$ yield lower scores. Figs. 11a and 11b together indicate that the optimal values for $\mathrm{LR_{CE}}$ and $\mathrm{LR_{other}}$ are not of the same order of

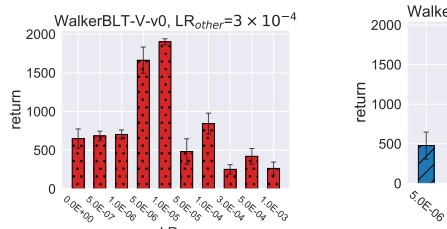 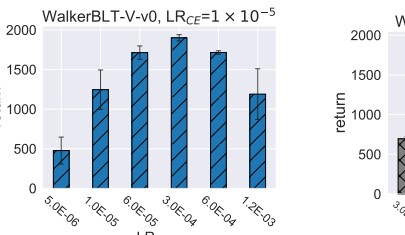 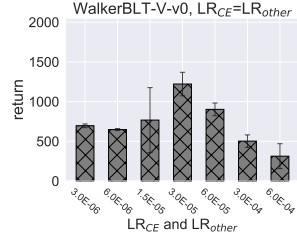

(a) Fixing $LR_{other}$, varying $LR_{CE}$. (b) Fixing $LR_{CE}$, varying $LR_{other}$. (c) $LR_{CE} = LR_{other}$, varying both.

Figure 11: Sensitivity studies of varied learning rates in terms of the average final return.

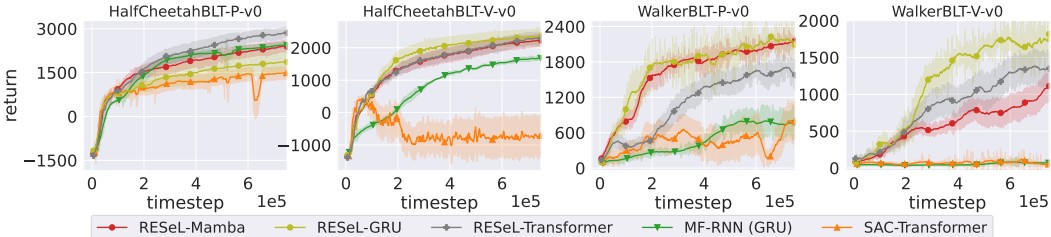

Figure 12: Learning curves shaded with 1 standard error with different RNN architectures.

magnitude. Additionally, it can also be found from Fig. 11c that setting $LR_{CE}$ equally significantly degrades policy performance. This emphasizes the importance of using different learning rates for context encoder and other layers. We extended the experiments from Figs. 11a and 11c to additional seven POMDP tasks, with results presented in Fig. 20 in Appendix E.6. The conclusions drawn from Fig. 20 are largely consistent with those from Figs. 11a and 11c, indicating that the impact of the context-encoder-specific learning rate on training is relatively generalizable.

**Ablation Studies.** We then explore whether the context-encoder-specific learning rate applies to other RNN or Transformer architectures [9]. We integrated a GRU [8] or Transformer as the context encoder while keeping all other settings constant. Due to the high computational cost of training GRUs and Transformers on full-length trajectories, we tested the RESeL variant with these architectures on four classic POMDP tasks for 0.75M time steps. Figure 12 displays the learning curves for various variants. MF-RNN [11] and SAC-Transformer [38] serve as baselines based on GRUs and Transformers from prior studies.

We observe that the context-encoder-specific learning rate also improves and stabilizes the performance of the GRU and Transformer variants. Our findings show that both RESeL-GRU and RESeL-Transformer perform comparably to RESeL-Mamba on three out of four tasks. These results suggest that the performance gains of RESeL are not solely attributed to an advanced RNN architecture, and that different arhitectures may be suited to different tasks. Additional sensitivity experiments and ablation studies on context length can be found in Appendices E.4 to E.6.

## 6 Conclusions and Limitations

In this paper, we proposed RESeL, an efficient recurrent off-policy RL algorithm. RESeL tackles the training stability issue existing in previous recurrent RL methods. RESeL uses difference learning rates for the RNN context encoder and other fully-connected layers to improve the training stability of RNN and ensure the performance of MLP. Experiments in various POMDP benchmarks showed that RESeL can achieve or surpass previous SOTA across a wider variety of tasks, including classic POMDP tasks, meta-RL tasks, and credit assignment tasks.

**Limitations.** We also notice that there are several noticeable limitations concerning this work: (1) We have shown that RESeL can even surpass SOTA methods in some MDP tasks, but it is unclear how the RNN helps the policy improve the performance. (2) RESeL introduces one more hyper-parameter, i.e., $LR_{CE}$. It is would be nice to explore the relationship between optimal $LR_{CE}$ and $LR_{other}$ or automatically parameter-tuning strategy for $LR_{CE}$.

## Acknowledgments and Disclosure of Funding

This work was supported by the Fundamental Research Program for Young Scholars (PhD Candidates) of the National Science Foundation of China (623B2049) and the Jiangsu Science Foundation (BK20243039). The authors thank the anonymous reviewers and Ms. Qianqian Cheng for their constructive feedback on this paper.

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

# A Proof Proposition 1

The following proof process is primarily based on the proof approach in [44, 45]. To the best of our knowledge, this is the first time the compounding error analysis method has been applied to study the properties of RNNs.

Let an RNN network $f^\theta(x_t, h_t) : \mathcal{X} \times \mathcal{H} \to \mathcal{Y} \times \mathcal{H}$ be formalized as:

$$y_t, h_{t+1} = f^\theta(x_t, h_t),$$

where $\theta$ is the network parameter, $x_t$ is the RNN input, $y_t$ is the RNN output, $h_t$ and $h_{t+1}$ are the RNN hidden states. For simplicity, we denote

$$y_t = f_y^\theta(x_t, h_t),$$
$$h_{t+1} = f_h^\theta(x_t, h_t).$$

We consider another RNN parameter $\theta'$, which is obtained by updating $\theta$ by one gradient step. We make the following assumptions:

1. **A1 (Lipschitz continuity):** $f^\theta$ and $f^{\theta'}$ both satisfy the Lipschitz continuity. There exist $K_h \in [0, 1)^3$ and $K_y \in \mathbb{R}$ satisfying:

$$\left\| f_h^\theta(x, h) - f_h^\theta(x, h') \right\| \le K_h \left\| h - h' \right\|, \quad \forall x \in \mathcal{X}, h, h' \in \mathcal{H}, \tag{A1.1}$$

$$\left\| f_h^{\theta'}(x, h) - f_h^{\theta'}(x, h') \right\| \le K_h \left\| h - h' \right\|, \quad \forall x \in \mathcal{X}, h, h' \in \mathcal{H}, \tag{A1.2}$$

$$\left\| f_y^\theta(x, h) - f_y^\theta(x, h') \right\| \le K_y \left\| h - h' \right\|, \quad \forall x \in \mathcal{X}, h, h' \in \mathcal{H}, \tag{A1.3}$$

$$\left\| f_y^{\theta'}(x, h) - f_y^{\theta'}(x, h') \right\| \le K_y \left\| h - h' \right\|, \quad \forall x \in \mathcal{X}, h, h' \in \mathcal{H}. \tag{A1.4}$$

2. **A2 (one-step output difference constraint):** The output differences of the RNN parameterized by $\theta$ and $\theta'$ are bounded by a constant $\epsilon \in \mathbb{R}$:

$$\left\| f_h^\theta(x, h) - f_h^{\theta'}(x, h) \right\| \le \epsilon, \quad \forall x \in \mathcal{X}, h \in \mathcal{H}, \tag{A2.1}$$

$$\left\| f_y^\theta(x, h) - f_y^{\theta'}(x, h) \right\| \le \epsilon, \quad \forall x \in \mathcal{X}, h \in \mathcal{H}. \tag{A2.2}$$

With the above assumptions, we now prove Proposition 1 in following.

*Proof.* $\forall x \in \mathcal{X}, h, h' \in \mathcal{H}$, we have

$$\left\| f_h^{\theta'}(x, h') - f_h^\theta(x, h) \right\|$$
$$= \left\| f_h^{\theta'}(x, h') - f_h^{\theta'}(x, h) + f_h^{\theta'}(x, h) - f_h^\theta(x, h) \right\|$$
$$\le \left\| f_h^{\theta'}(x, h') - f_h^{\theta'}(x, h) \right\| + \left\| f_h^{\theta'}(x, h) - f_h^\theta(x, h) \right\| \tag{2}$$
$$\overset{\text{Eq. (A1.2)}}{\le} K_h \left\| h - h' \right\| + \left\| f_h^{\theta'}(x, h) - f_h^\theta(x, h) \right\|$$
$$\overset{\text{Eq. (A2.1)}}{\le} K_h \left\| h - h' \right\| + \epsilon.$$

Similarly, $\forall x \in \mathcal{X}, h, h' \in \mathcal{H}$, we also have

$$\left\| f_y^{\theta'}(x, h') - f_y^\theta(x, h) \right\| \le K_y \left\| h - h' \right\| + \epsilon. \tag{3}$$

Given an input sequence $\{x_0, x_1, \ldots, x_T\}$ with sequence length $T + 1$ and an initial hidden state $h_0$, let the output and hidden state sequences produced by $f^\theta$ be $\{y_0, y_1, \ldots, y_T\}$ and $\{h_1, \ldots, h_{T+1}\}$.

---

[3]We only consider $K_h \in [0, 1)$ as most RNN models satisfy this condition. GRU/LSTM and SSMs achieve this condition by adopting a sigmoid activation function and directly limits the value of the transition matrix of the hidden state, respectively. The RNN with a $K_h > 1$ could diverge during long-sequence forwarding.

Similarly, let the output and hidden state sequences produced by $f^{\theta'}$ be $\{y'_0, y'_1, \ldots, y'_T\}$ and $\{h'_1, \ldots, h'_{T+1}\}$. We can measure the upper bound of the hidden state differences between $f^\theta$ and $f^{\theta'}$.

$$\|h_1 - h'_1\| = \left\| f_h^\theta(x_0, h_0) - f_h^{\theta'}(x_0, h_0) \right\| \overset{\text{Eq. (A2.1)}}{\leq} \epsilon. \tag{4}$$

$$\begin{aligned}
\|h_2 - h'_2\| &= \left\| f_h^\theta(x_1, h_1) - f_h^{\theta'}(x_1, h'_1) \right\| \\
&\overset{\text{Eq. (2)}}{\leq} K_h \|h_1 - h'_1\| + \epsilon \\
&\overset{\text{Eq. (4)}}{\leq} K_h \epsilon + \epsilon.
\end{aligned} \tag{5}$$

Similarly, we have

$$\|h_3 - h'_3\| = \left\| f_h^\theta(x_2, h_2) - f_h^{\theta'}(x_2, h'_2) \right\| \leq \sum_{i=0}^{2} K_h^i \epsilon. \tag{6}$$

$$\|h_t - h'_t\| = \left\| f_h^\theta(x_{t-1}, h_{t-1}) - f_h^{\theta'}(x_{t-1}, h'_{t-1}) \right\| \leq \sum_{i=0}^{t-1} K_h^i \epsilon. \tag{7}$$

Then we can derive the output difference at the $t$-th step ($t > 0$):

$$\begin{aligned}
\|y_t - y'_t\| &= \left\| f_y^\theta(x_t, h_t) - f_y^{\theta'}(x_t, h'_t) \right\| \\
&\overset{\text{Eq. (3)}}{\leq} K_y \|h_t - h'_t\| + \epsilon \\
&\overset{\text{Eq. (7)}}{\leq} K_y \sum_{i=0}^{t-1} K_h^i \epsilon + \epsilon.
\end{aligned} \tag{8}$$

As $0 \leq K_h < 1$, Eq. (8) can be further simplified to

$$\|y_t - y'_t\| \leq K_y \frac{1 - K_h^t}{1 - K_h} \epsilon + \epsilon. \tag{9}$$

Eq. (9) also holds for $t = 0$, i.e., $\|y_0 - y'_0\| \leq \epsilon$. Thus, we finally have

$$\|y_t - y'_t\| \leq K_y \frac{1 - K_h^t}{1 - K_h} \epsilon + \epsilon, \quad \forall t \geq 0. \tag{10}$$

$\square$

## B   Average Output Differences Bound

**Proposition 2.** *Following the setting in Proposition 1, the average output difference satisfies*

$$\frac{1}{t+1} \sum_{i=0}^{t} \|y_t - y'_t\| \leq \frac{K_y}{1 - K_h} \epsilon + \epsilon - K_y \frac{1 - K_h^{t+1}}{(1+t)(1 - K_h)^2} \epsilon, \quad \forall t \geq 0. \tag{11}$$

*Proof.* From Eq. (10), we have

$$\|y_t - y'_t\| \leq \frac{K_y}{1 - K_h} \epsilon + \epsilon - K_y \frac{\epsilon}{1 - K_h} K_h^t.$$

As

$$\frac{1}{t+1} \sum_{i=0}^{t} K_h^i$$

$$= \frac{1 - K_h^{t+1}}{(1 - K_h)(t+1)},$$

**Algorithm 1:** Training Procedure of RESeL

---

**Input:** Environment $E$; Policy $\pi_\phi$; Ensemble critic $Q_\psi$; Target ensemble critic $Q_{\psi'}$; Batch size $BS$; Entropy coefficient $\alpha$; Target entropy $TE$;

1   Initialize context-encoder-specific policy optimizer and critic optimizer;
2   Initialize an empty replay buffer $\mathcal{B} \leftarrow \emptyset$ and trajectory $\tau \leftarrow \emptyset$;
3   Initial context $\hat{a} \leftarrow 0, \hat{o} \leftarrow 0, \hat{r} \leftarrow 0$;
4   Initial hidden state $h \leftarrow 0$;
5   $t \leftarrow 0$;
6   Obtain initial observation $o_0$ from $E$;
7   **while** $t < max\_step$ **do**
8      $a_t, h \leftarrow \pi_\phi(o_t, \hat{o}, \hat{a}, \hat{r}, h)$;
9      Execute $a_t$ to $E$;
10      Observe new observation $o_{t+1}$, reward $r_t$, and terminal signal $d_t$ ;
11      Add $(o_t, a_t, r_t, o_{t+1})$ to $\tau$;
12      Update context data $\hat{a} \leftarrow a_t, \hat{o} \leftarrow o_t, \hat{r} \leftarrow r_t$;
13      **if** $d_t$ *is True* **then**
14          Insert $\tau$ to $\mathcal{B}$;
15          $\tau \leftarrow \emptyset$;
16          Reset context $\hat{a} \leftarrow 0, \hat{o} \leftarrow 0, \hat{r} \leftarrow 0$;
17          Reset hidden state $h \leftarrow 0$;
18          Reset $E$, obtain new observation $o_{t+1}$ from $E$;
19      Sample data batch $D = \{\tau_i \mid i = 1, 2, \dots\}$ with data count $\sum_i |\tau_i| \geq BS$ from $\mathcal{B}$;
20      Obtain REDQ [21] critic loss with $Q_\psi$, $Q_{\psi'}$, $\alpha$, and $D$ following Eq. (19);
21      Update $\psi$ with the critic optimizer via one-step gradient descent;
22      Softly assign $\psi$ to $\psi'$;
23      **if** $t\%2 = 0$ **then**
24          Obtain REDQ [21] policy loss with $\pi_\phi$, $\alpha$, and $D$ following Eq. (20);
25          Update $\phi$ with the policy optimizer via one-step gradient descent;
26          Automatically tune $\alpha$ to control the policy entropy to $TE$;
27      $t \leftarrow t + 1$;

---

we can get

$$\frac{1}{t+1} \sum_{i=0}^{t} \|y_i - y_i'\|$$

$$\leq \frac{K_y}{1 - K_h}\epsilon + \epsilon - K_y \frac{\epsilon}{1 - K_h} \frac{1}{t+1} \sum_{i=0}^{t} K_h^t$$

$$= \frac{K_y}{1 - K_h}\epsilon + \epsilon - K_y \frac{1 - K_h^{t+1}}{(1 - K_h)^2(t+1)}\epsilon.$$

$\square$

## C   Algorithmic Details

### C.1   Detailed Procedure of RESeL

The training procedure of RESeL is summarized in Algorithm 1. In the following part, we elaborate on the form of REDQ [21] losses in the setting of recurrent RL. Let the critic $Q_\psi(o, a, \hat{o}, \hat{a}, \hat{r}, h_Q)$ : $\mathcal{O} \times \mathcal{A} \times \mathcal{O} \times \mathcal{A} \times \mathbb{R} \rightarrow \mathbb{R}^8 \times \mathcal{H}$ be a mapping function from the current observation $o$ and action $a$, last-step observation $\hat{o}$, action $\hat{a}$, reward $\hat{r}$, and hidden state $h_Q$ to a 8-dim Q-value vector and the next hidden state $h_Q$. The target critic $Q_{\psi'}(o, a, \hat{o}, \hat{a}, \hat{r}, h_Q)$ shares the same architecture to $Q_\psi$ but parameterized by $\psi'$. The policy $\pi(o, \hat{o}, \hat{a}, \hat{r}, h_\pi)$ : $\mathcal{O} \times \mathcal{A} \times \mathcal{O} \times \mathcal{A} \times \mathbb{R} \rightarrow \Delta_\mathcal{A} \times \mathcal{H}$ is a mapping function from the current observation $o$ , last-step observation $\hat{o}$, action $\hat{a}$, reward $\hat{r}$, and hidden state $h_\pi$ to an action distribution and the next hidden state $h_\pi$.

Given a batch of data $D = \{\tau_i \mid i = 1, 2, ...\}$, where $\tau_i = \{o_0^i, a_0^i, r_0^i, o_1^i, a_1^i, r_1^i, o_2^i, \ldots, r_{L_i-1}^i, o_{L_i}^i\}$ is the $i$-th trajectory in the batch, $L_i$ is the trajectory length. We then re-formalize $\tau_i$ to tensorized data: $\mathbf{o}_i \in \mathcal{O}^{L_i}, \mathbf{a}_i \in \mathcal{A}^{L_i}, \hat{\mathbf{o}}_i \in \mathcal{O}^{L_i}, \hat{\mathbf{a}}_i \in \mathcal{A}^{L_i}, \mathbf{r}_i \in \mathbb{R}^{L_i}, \hat{\mathbf{r}}_i \in \mathbb{R}^{L_i}$, where, for example, $\mathbf{o}_i^t \in \mathcal{O} = o_t^i$ is the $t$-th row of $\mathbf{o}_i$. $\hat{\mathbf{o}}_i^0 = \mathbf{0}, \hat{\mathbf{a}}_i^0 = \mathbf{0}, \hat{\mathbf{r}}_i^0 = 0$ as the there is no last step at the first time step. The policy receives the tensorized data and outputs action distributions. For simplicity, we formalize the outputs of the policy as the action sampled from the action distribution and its logarithm sampling probability:

$$\tilde{\mathbf{a}}_i, \log \tilde{\mathbf{p}}_i, h_\pi^{L_i} \leftarrow \pi_\phi(\mathbf{o}_i, \hat{\mathbf{o}}_i, \hat{\mathbf{a}}_i, \hat{\mathbf{r}}_i, \mathbf{0}), \tag{12}$$

where $\tilde{\mathbf{a}}_i \in \mathcal{A}^{L_i}, \log \tilde{\mathbf{p}}_i \in \mathbb{R}^{L_i}, h_\pi^{L_i} \in \mathcal{H}$ are obtained autoregressively:

$$
\begin{aligned}
\tilde{\mathbf{a}}_i^0, \log \tilde{\mathbf{p}}_i^0, h_\pi^1 &\leftarrow \pi_\phi(\mathbf{o}_i^0, \hat{\mathbf{o}}_i^0, \hat{\mathbf{a}}_i^0, \hat{\mathbf{r}}_i^0, \mathbf{0}), \\
\tilde{\mathbf{a}}_i^1, \log \tilde{\mathbf{p}}_i^1, h_\pi^2 &\leftarrow \pi_\phi(\mathbf{o}_i^1, \hat{\mathbf{o}}_i^1, \hat{\mathbf{a}}_i^1, \hat{\mathbf{r}}_i^1, h_\pi^1), \\
&\cdots\cdots \\
\tilde{\mathbf{a}}_i^t, \log \tilde{\mathbf{p}}_i^t, h_\pi^{t+1} &\leftarrow \pi_\phi(\mathbf{o}_i^t, \hat{\mathbf{o}}_i^t, \hat{\mathbf{a}}_i^t, \hat{\mathbf{r}}_i^t, h_\pi^t).
\end{aligned}
\tag{13}
$$

Similarly, the critic model receives the tensorized data and outputs 8-dim Q-values as follows:

$$\tilde{\mathbf{Q}}_i, h_Q^{L_i} \leftarrow Q_\psi(\mathbf{o}_i, \mathbf{a}_i, \hat{\mathbf{o}}_i, \hat{\mathbf{a}}_i, \hat{\mathbf{r}}_i, \mathbf{0}), \tag{14}$$

where $\tilde{\mathbf{Q}}_i \in \mathbb{R}^{L_i \times 8}, h_\pi^{L_i} \in \mathcal{H}$ are obtained autoregressively:

$$
\begin{aligned}
\mathbf{Q}_i^0, h_Q^1 &\leftarrow Q_\psi(\mathbf{o}_i^0, \mathbf{a}_i^0, \hat{\mathbf{o}}_i^0, \hat{\mathbf{a}}_i^0, \hat{\mathbf{r}}_i^0, \mathbf{0}), \\
\mathbf{Q}_i^1, h_Q^2 &\leftarrow Q_\psi(\mathbf{o}_i^1, \mathbf{a}_i^1, \hat{\mathbf{o}}_i^1, \hat{\mathbf{a}}_i^1, \hat{\mathbf{r}}_i^1, h_Q^1), \\
&\cdots\cdots \\
\mathbf{Q}_i^t, h_Q^{t+1} &\leftarrow Q_\psi(\mathbf{o}_i^t, \mathbf{a}_i^t, \hat{\mathbf{o}}_i^t, \hat{\mathbf{a}}_i^t, \hat{\mathbf{r}}_i^t, h_Q^t).
\end{aligned}
\tag{15}
$$

Note that, in Mamba, the process demonstrated in Eq. (13) and Eq. (15) can be parallelized.

In order to compute the target value, we first extend the tensorized data

$$
\begin{aligned}
\mathbf{o}_i^+ &\leftarrow \{\mathbf{o}_i, \mathbf{o}_i^{L_i}\}, \\
\hat{\mathbf{o}}_i^+ &\leftarrow \{\hat{\mathbf{o}}_i, \mathbf{o}_i^{L_i-1}\}, \\
\hat{\mathbf{a}}_i^+ &\leftarrow \{\hat{\mathbf{a}}_i, \mathbf{a}_i^{L_i-1}\}, \\
\hat{\mathbf{r}}_i^+ &\leftarrow \{\hat{\mathbf{r}}_i, \mathbf{r}_i^{L_i-1}\}, \\
\mathbf{r}_i^+ &\leftarrow \{0, \mathbf{r}_i\},
\end{aligned}
\tag{16}
$$

where $\mathbf{o}_i^{L_i}$ is the next observation of the last step, $\mathbf{o}_i^{L_i-1}, \mathbf{a}_i^{L_i-1}$, and $\mathbf{r}_i^{L_i-1}$ are the observation, action, and reward at the last step, respectively.

The target value can be obtained:

$$
\begin{aligned}
\tilde{\mathbf{a}}_{\text{target},i}^+, \log \tilde{\mathbf{p}}_{\text{target},i}, h_\pi^{L_i+1} &\leftarrow \pi_\phi(\mathbf{o}_i^+, \hat{\mathbf{o}}_i^+, \hat{\mathbf{a}}_i^+, \hat{\mathbf{r}}_i^+, \mathbf{0}), \\
\tilde{\mathbf{Q}}_{\text{target},i}^+, h_Q^{L_i+1} &\leftarrow \mathbf{r}_i^+ + \gamma \left( Q_{\psi'} \left( \mathbf{o}_i^+, \tilde{\mathbf{a}}_{\text{target},i}^+, \hat{\mathbf{o}}_i^+, \hat{\mathbf{a}}_i^+, \hat{\mathbf{r}}_i^+, \mathbf{0} \right) - \alpha \log \tilde{\mathbf{p}}_{\text{target},i} \right).
\end{aligned}
\tag{17}
$$

Then, we sample a set $\mathcal{M}$ of 2 distinct indices from $\{0, 1, ..., 7\}$. Let $\tilde{\mathbf{Q}}_{\text{target},i}^{+,j}$ be the $j$-th column of $\tilde{\mathbf{Q}}_{\text{target},i}^+$. We choose the minimum value from $\{\tilde{\mathbf{Q}}_{\text{target},i}^{+,j} | j \in \mathcal{M}\}$:

$$\mathbf{Q}_{\text{target},i}^+ = \min_{j \in \mathcal{M}} \tilde{\mathbf{Q}}_{\text{target},i}^{+,j}, \tag{18}$$

where $\mathbf{Q}_{\text{target},i}^+ \in \mathbb{R}^{L_i+1}$. We then let $y_i$ be the last $L_i$ rows of $\mathbf{Q}_{\text{target},i}^+$:

$$y_i = \mathbf{Q}_{\text{target},i}^+[1 : L_i + 1].$$

The REDQ critic loss can be written as

$$\mathcal{L}_Q = \frac{1}{\sum_{i=1}^{|D|} L_i} \sum_{i=1}^{|D|} \left\| y_i - \tilde{\mathbf{Q}}_i \right\|_2^2, \tag{19}$$

where $\tilde{\mathbf{Q}}_i$ is obtained following Eq. (14), $\| \cdot \|_2$ denotes L2-norm. As for the policy loss, we first obtain $\tilde{\mathbf{a}}_i$ and $\log \tilde{\mathbf{p}}_i$ via Eq. (12). We can get the critic value $\tilde{\mathbf{Q}}_i^\star$ corresponding to the new action $\tilde{\mathbf{a}}_i$ following Eq. (14):

$$\tilde{\mathbf{Q}}_i^\star, h_Q^{L_i} \leftarrow Q_\psi(\mathbf{o}_i, \tilde{\mathbf{a}}_i, \hat{\mathbf{o}}_i, \hat{\mathbf{a}}_i, \hat{\mathbf{r}}_i, \mathbf{0}).$$

We can get the policy loss:

$$\mathcal{L}_\pi = -\frac{1}{8\sum_{i=1}^{|D|} L_i} \text{sum}\left(\tilde{\mathbf{Q}}_i^\star - \alpha \log \tilde{\mathbf{p}}\right), \tag{20}$$

where denominator of 8 results from our ensemble size being 8, requiring the averaging of the outputs of the eight critics to compute the policy loss.

## C.2 Sample Stacked Batch from Replay Buffer

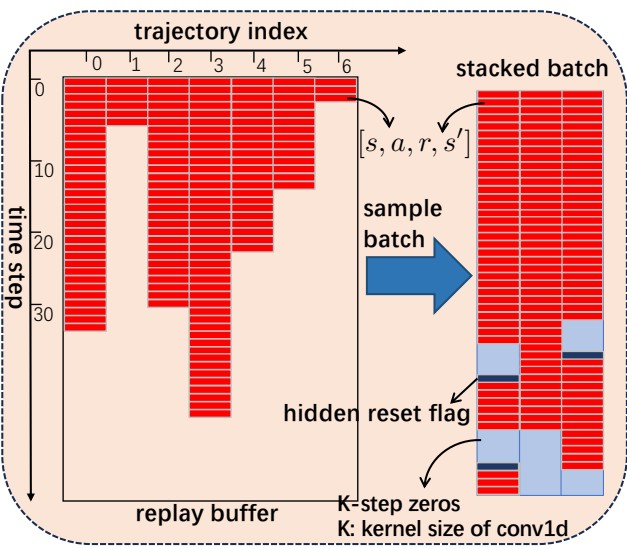

Figure 13: Illustrating for sampling a stacked batch from replay buffer.

In RESeL, we collect several full-length trajectories from the replay buffer as a batch for policy training. However, in many environments, trajectory lengths could vary a lot, making it impossible to directly concatenate them into a single tensor for subsequent computations. Typically, we pad trajectories to the same length with zeros, resulting in a tensor of shape $[B, L, C]$, where $B$ is the batch size, $L$ is the sequence length, and $C$ is the tensor dimension. Additionally, we introduce a mask tensor of shape $[B, L, 1]$, where elements equal to 1 indicate valid data, and 0s indicate padding. We then use a modified mean operation:

$$mask\_mean(x, mask) = \sum(x \times mask)/sum(mask),$$

to prevent padding from biasing the mean. This approach allows for GPU parallelization of all trajectories in the batch, though calculations for padded 0s are redundant. If trajectory lengths vary greatly, the proportion of padding can become significant, severely reducing computational efficiency, even resulting in out-of-GPU-memory. For instance, in a batch with one trajectory of length 1000 and ten trajectories of length 50, there would be 1500 valid data points and 9500 padded points, substantially lowering efficiency.

To address this issue, we adopt the batch sampling method illustrated in Fig. 13, stacking shorter trajectories along the time dimension. This requires the RNN to accept a hidden reset flag, which we insert at the beginning of each trajectory to reset the hidden state. Additionally, we insert $K$ steps of zero data between trajectories to prevent convolution1d from mixing adjacent trajectory data, as shown in Fig. 13. To implement this sampling strategy, we modify the original Mamba implementation[4], adding an input denoting hidden reset flag. When occurring the hidden reset flag

---

[4] https://github.com/state-spaces/mamba

with value of 1, the hidden state will be reset to zero. The introduction of hidden reset flag will not influence the parallelizability of Mamba but can avoid potential redundant computation.

# D  Experiment Details

## D.1  Descriptions of the Environments

### D.1.1  Classic POMDP Tasks

These classic POMDP tasks[5] are developed using PyBullet locomotion environments. The environments include AntBLT, HopperBLT, WalkerBLT, and HalfCheetahBLT. Partial observability is introduced by obscuring parts of the observation. For each environment, two types of tasks are created: one where only joint velocity is observable and another where only joint position is observable. We denote these as "-V" for joint velocity information and "-P" for joint position information. In such settings, the agent must reconstruct the missing information, whether velocity or position, based on historical observations.

### D.1.2  Dynamics-Randomized Tasks

The Dynamics-Randomized Tasks[6] environments are based on the MuJoCo environment [37], following the training settings in [16]. In these tasks, the environment's gravity is altered using an open-source tool[7]. The gravity is sampled as following procedure: We first sample a value $a \in \mathbb{R}$ uniformly from $[-3, 3]$. Let the original gravity in MuJoCo be $g_{\text{origin}}$, the new gravity is obtained as:

$$g_{\text{new}} = 1.5^a g_{\text{origin}}. \tag{21}$$

Before training, we sample 60 values of $g_{\text{new}}$. Of these, 40 are used for training and 20 for testing the learned policies. In these tasks, the agent must infer the gravity based on historical interactions with the environment and adjust its policy accordingly.

### D.1.3  Classic meta-RL Tasks

These meta-RL tasks[8] have been utilized in various meta-RL algorithms [42, 19]. Specifically, in these tasks, the reward function is modified in the Ant and HalfCheetah MuJoCo environments. In the 'Vel' tasks, the agent is rewarded based on its proximity to a desired velocity. In the 'Dir' tasks, the agent must move in a desired direction. The desired velocity and direction information are reflected only in the reward function. In the Wind environment, an external, unobservable disturbance factor (wind speed) influences state transitions. The agent must navigate to a goal under varying wind conditions. In these tasks, the agent uses past interaction data, including rewards, to infer the underlying reward functions or transition disturbance factors.

### D.1.4  Key-to-Door

The Key-to-Door environment[9] consists of three phases. In the first phase, the agent spawns at a random location on the map, and a key will appear at another location on the map, the agent can touch the key. In the second phase, the agent will have the task of picking up apples. In the third phase, if the agent has touched the key in the first phase, it can receive a reward; otherwise, it cannot.

The action space is four-dimensional, encompassing the actions of moving forward, backward, left and right. The reward structure is dual-faceted: it includes rewards garnered from the collection of apples during the second phase and the ultimate reward obtained upon door opening in the final phase. The transition function is defined by the positional changes of the agent subsequent to the execution of any of the four actions, with stage transitions dictated by an intrinsic timer, which facilitates a shift to the subsequent stage upon the expiration of the countdown. The optimal policy should locate and touch the key in the initial phase, which is served for the final reward acquisition upon door opening

---

[5]https://github.com/oist-cnru/Variational-Recurrent-Models
[6]https://github.com/FanmingL/ESCP
[7]https://github.com/dennisl88/rand_param_envs
[8]https://github.com/lmzintgraf/varibad
[9]https://github.com/twni2016/Memory-RL

in the third phase, while maximizing apple collection during the second phase. The credit assignment length in Fig. 10 denotes the time period staying at the second stage.

## D.2 Baselines

In the section, we give brief introductions of the baseline algorithms we comparing in our work. We first introduce the baselines in the classic POMDP tasks Fig. 7.

- *Variational Recurrent Model* (VRM) [18]. VRM utilizes a variational RNN [46] to learn a transition model, and feeding the RNN hidden state into an MLP policy to obtain the action.

- *GPIDE* (GPIDE-ESS) [38]. Inspired by the PID controller's success, which relies on summing and derivative to accumulate information over time, GPIDE propose two history encoding architectures: one using direct PID features and another extending these principles for general control tasks by consisting of a number of "heads" for accumulating information about the history.

- *Recurrent Model-free RL* (MF-RNN) [11]. MF-RNN proposes several design considerations in recurrent off-policy RL methods. MF-RNN analyzed the best choice concerning the context length, RL algorithm, whether sharing the RNN between critic and policy networks, etc. We used the best results reported in [11] in Fig. 7.

The baseline results in Fig. 7 come from two sources: (1) The evaluation results provided in [11][10], including MF-RNN, PPO-GRU, SAC-MLP, TD3-MLP, VRM, and A2C-GRU; (2) The results provided in [38][11], consisting GPIDE-ESS and SAC-Transformer.

The baselines compared in Fig. 8 are elaborated as follows:

- *Environment Probing Interaction Policy* (EPI) [17]. This baseline aims to extracting features from the transition. A context encoder is obtained to enable a transition model to predict the future state through minimizing the following loss:

$$\mathcal{L}_{\text{CE}}^{\text{DM}} = \mathbb{E}_{z_i^t, s_t, a_t, s_{t+1}} \left\| T(s_t, a_t, z_i^t; \theta_T) + s_t - s_{t+1} \right\|_2^2, \tag{22}$$

where $T(s_t, a_t, z_i^t; \theta_T)$ is a transition model parameterized by $\theta_T$, $z_i^t$ is the output of the context encoder. In addition, the transition model is also optimized as described in Eq. (22).

- *Online system identification* (OSI) [33]. OSI is structured by a context encoder which is used to predict the parameters of the environment dynamics. The architecture of OSI is the same as Fig. 1. OSI adopts a supervised learning loss function to train the context encoder: $\mathcal{L}_{\text{CE}}^{\text{OSI}} = \mathbb{E}_{z_i^t} \|z_i^t - c_i\|_2^2$, where $c_i$ is the dynamics parameter such as `gravity`, $z_i^t$ is the embedded context.

- *Environment sensitive contextual policy learning* (ESCP) [16][12]. This approach is design to improve both the the robustness and sensitivity context encoder, based on a contrastive loss. In order to improve the policy robustness in environment where dynamics sudden changes could occurs, ESCP truncate the memory length in both training and deployment phase.

- *Proximal Meta-Policy Search* (ProMP) [40][13]. This method learns a pretrained policy model in training phase and finetune it in deployment phase with few gradient steps. The reported performance of ProMP is obtained by the policies after finetuning.

- *Soft actor-critic with MLP network* (SAC-MLP) [20]. An implementation of the soft actor-critic algorithm with MLP policy and critic models.

- *Probabilistic embeddings for actor-critic RL* (PEARL) [42]. The context encoder of PEARL is trained to make a precise prediction of the action-value function while maximize entropy. The loss of the context encoder can be written as:

$$\mathcal{L}_{\text{CE}}^{\text{PEARL}} = \mathcal{L}_{\text{critic}} + \text{D}_{\text{KL}}[p(z \mid \tau) \| \mathcal{N}(0, I)]$$

---

[10]`https://github.com/twni2016/pomdp-baselines`
[11]`https://github.com/ianchar/gpide`
[12]`https://github.com/FanmingL/ESCP`
[13]`https://github.com/jonasrothfuss/ProMP`

where $\mathcal{L}_{\text{critic}}$ is the critic loss in SAC. $D_{\text{KL}}[p(z|\tau) \parallel \mathcal{N}(0, I)]$ denotes the KL divergence between distribution of the embedded context and a standard Gaussian distribution. In $p(z|\tau)$, $z$ denotes the output of the context encoder based on a given trajectory. PEARL is implemented with the officially provided codes[14].

- *SAC-GRU.* This is an upgrade version of SAC with GRU models. It use standard SAC algorithm to optimize an RNN policy.

The baselines compared in Fig. 9:

- *RL$^2$* [25]. RL$^2$ is a meta-RL algorithm that trains a meta-learner to efficiently adapt to new tasks by leveraging past experiences. It involves two nested loops: an inner loop where the meta-learner interacts with the environment to learn task-specific policies, and an outer loop where the meta-learner updates its parameters based on the performance across multiple tasks. The inner loop is accomplished by an RNN.

- *VariBad* [19]. VariBAD (Variational Bayes for Adaptive Deep RL) is also a meta-RL method that uses variational inference to enable fast adaptation to new tasks. It employs a Bayesian approach to meta-learning, inferring task-specific posterior distributions over the parameters of the policy and value function. Specifically, it first learns an RNN variational auto-encoder to extract the sequential features of the dynamics or tasks. The policy then receives the current state and the output of the context encoder to make adaptive decisions.

The results of the baselines in Fig. 9 are sourced from [11].

MF-GPT [15], i.e. the baseline in Fig. 10, denotes to train a Transformer network, i.e. GPT-2, to solve the memory-based tasks. The baseline result of the baselines in Fig. 10 are from [15][15].

### D.3    Implementation Details

#### D.3.1    State Space Models

In our implementation, we use Mamba to form the context encoder, as it supports processing sequences in parallel. Therefore, we will first introduce state space models and Mamba before discussing the network architecture.

**State Space Model (SSM)** is a sequential model [47, 14, 48] commonly used to model dynamic systems. The discrete version of an SSM can be expressed as:

$$
\begin{aligned}
h_t &= \overline{\mathbf{A}} h_{t-1} + \overline{\mathbf{B}} x_t, \quad y_t = \mathbf{C} h_t, \\
\overline{\mathbf{A}} &= \exp(\Delta \mathbf{A}), \quad \overline{\mathbf{B}} = (\Delta \mathbf{A})^{-1}(\exp(\Delta \mathbf{A}) - I) \cdot \Delta \mathbf{B},
\end{aligned}
\tag{23}
$$

where $\Delta$, $\mathbf{A}$, $\mathbf{B}$, and $\mathbf{C}$ are parameters; $x_t$, $y_t$, and $h_t$ represent the input, output, and hidden state at time step $t$, respectively. $\overline{\mathbf{A}}$ and $\overline{\mathbf{B}}$ are derived from $\Delta$, $\mathbf{A}$, and $\mathbf{B}$.

Mamba [14] is an RNN variant that integrates SSM layers to capture temporal information. In Mamba, the parameters $\Delta$, $\mathbf{B}$, and $\mathbf{C}$ are dependent on the input $x_t$, forming an input-dependent selection mechanism. Unlike traditional RNNs which repeatedly apply Eq. (23) in a for-loop to generate the output sequence, Mamba employs a parallel associative scan, which enables the computation of the output sequence in parallel. This method significantly enhances computational efficiency.

#### D.3.2    Network Architecture

**Pre-encoder.** The pre-encoders consist of one-layer MLPs with a consistent output dimension of 128.

**Context Encoder.** The context encoder is comprised of one or more Mamba blocks, each flanked by two MLPs. Typically, the MLP before the Mamba block consists of a single layer with 256 neurons, except for the Key-to-Door task, where a three-layer MLP with 256 neurons per layer is employed.

Except in Key-to-Door task, a single Mamba block is utilized, with a state dimension of 64. The kernel size of the convolution1d layer is fixed at 8. Additionally, for these tasks, a position-wise

---

[14]https://github.com/katerakelly/oyster
[15]https://github.com/twni2016/Memory-RL

feedforward network, used by Transformer [9], is appended to the end of the Mamba block to slightly enhance stability during training. In Key-to-Door tasks, the convolution1d layer in Mamba is omitted, as the focus shifts to long-term credit assignment, rendering the convolution1d unnecessary for capturing short-term sequence features. Conversely, in Key-to-Door tasks with a credit assignment length of 500, four Mamba blocks are employed to improve the long-term memory capacity.

The MLP after the Mamba block also comprises a single layer with an output dimension of 128.

**MLP Policy/Critic.** As detailed in Sec. 4.1, both the MLP policy and critic consist of MLPs with two hidden layers, each comprising 256 neurons.

**Activations.** Linear activation functions are employed for the output of the pre-encoder, context encoder, and the MLP policy/critic. For other layers, the ELU activation function [49] is used.

**Optimizer.** We use AdamW [50] for RESeL training.

### D.3.3   Hyperparameters

The hyperparameters used for RESeL is listed in Table 2. We mainly tuned the learning rates, batch size for each tasks. $\gamma$ and last reward as input are determined according to the characteristic of the tasks. The batch size in Table 2 (1000 or 2000) is much larger than that used in previous MLP-based SAC algorithms (128 or 256) [20]. This is because we train on at least one complete trajectory at a time, and in most environments, the maximum trajectory length is 1000. Thus, batch sizes of 1000 or 2000 correspond to one or two trajectories of maximum length.

Table 2: Hyperparameters of RESeL.

| Attribute | Value | Task |
|---|:---:|---:|
| context encoder learning rate $\text{LR}_{\text{CE}}$ | $2 \times 10^{-6}$ 
 $10^{-5}$ | classic MuJoCo and classic meta-RL tasks 
 other tasks |
| other learning rate $\text{LR}_{\text{other}}$ for policy | $6 \times 10^{-5}$ 
 $3 \times 10^{-4}$ | classic MuJoCo and classic meta-RL tasks 
 other tasks |
| other learning rate $\text{LR}_{\text{other}}$ for value | $2 \times 10^{-4}$ 
 $10^{-3}$ | classic MuJoCo and classic meta-RL tasks 
 other tasks |
| $\gamma$ | 0.9999 
 0.99 | Key-to-Door 
 other tasks |
| last reward as input | True 
 False | classic meta-RL tasks 
 other tasks |
| batch size | 2000 
 1000 | classic POMDP tasks 
 other tasks |
| target entropy | $-1\times$ Dimension of action | all tasks |
| learning rate of $\alpha$ | $10^{-4}$ | all tasks |
| soft-update factor for target value network | 0.995 | all tasks |
| number of the randomly sampled data | 5000 | all tasks |

**Hyperparameter Determination** Some hyper-parameters (discount factor, reward as input) were chosen based on the nature of the environments, while others (Context encoder LR, policy and value function LR, and batch size) are determined via grid search:

- **Context Encoder LR:** We fixed the learning rate of the context encoder to be 1/30 of the policy learning rate in all tasks. We aim to have the action variation caused by RNN updates to be comparable to that caused by MLP updates, thus fixing the learning rate ratio between the context encoder and MLP. After conducting parameter searches in several environments, as shown in Fig. 20, we find the 1/30 ratio performed well, so we fixed this ratio across all environments.

- **Policy and Value Function LR:** We mainly referred to the default learning rate settings in CleanRL [51], using a policy learning rate of $3 \times 10^{-4}$ and an action-value function learning rate of $10^{-3}$. We conducted a grid search between the aforementioned learning rates and those reduced by a factor of 5 (aiming to further stabilize training). We found that a smaller

learning rate was more stable in tasks where the partial observability is not very significant (e.g., MDPs like classic MuJoCo).

- **Discount Factor ($\gamma$):** For general tasks, we used a common $\gamma$ value of 0.99. For environments requiring long-term memory (e.g., Key-to-Door), we used a larger $\gamma$ of 0.9999.

- **Reward as Input:** The Classic Meta-RL tasks require inferring the true reward function based on real-time rewards. Thus, we also included real-time reward feedback as an input to the policy and value in these tasks.

- **Batch Size:** We conducted a grid search for batch size, mainly exploring sizes that were 1x or 2x the maximum trajectory length. We found that in classic POMDP tasks, a larger batch size performed better. In other tasks, the advantage of a large batch size was not significant and could even reduce training speed.

# E  More Experimental Results

## E.1  Visualization

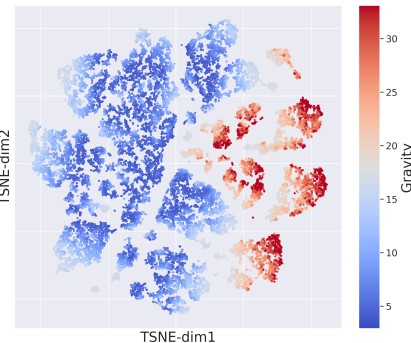

Figure 14: The t-SNE visualization of the outputs of the context encoder in Halfcheetah with different gravity.

In order to explore what the context encoder has learned, we attempt to conduct in-depth analysis on the embedded context data. We conducted experiments in the environment of Halfcheetah with varying gravity. We uniformly sample gravity acceleration within the range of $[2.90, 33.10] m/s^2$, resulting in 40 environments with different gravities. For each environment, we used a policy learned by RESeL to collect one trajectory, with a length of 1000 steps. Ultimately, we obtained data of size 40×1000, where each step's output of the encoder, a 128-dimensional vector, was saved. Subsequently, we performed t-SNE [52] processing on the 40×1000×128 data to map the high-dimensional 128-dimensional data into 2-dimensional data which can assist in our visualization analysis.

Finally, we presented the results in Fig. 14. In this figure, the colorbar represents the gravity acceleration magnitude. The $x$ and $y$ axes of the figure represent the two-dimensional data results after t-SNE mapping. From the figure, we can distinctly observe the non-random distribution characteristics of colors from left to right. The different data represented by the cool and warm colors are not randomly intermingled but rather exhibit certain clustering characteristics. We can find that the embeddings are strongly correlated with gravity acceleration, indicating that RESeL has recovered the hidden factor of the POMDP environment from the historical interaction data.

## E.2  Time Overhead Comparisons

We find the computational efficiency is also a common issue among previous recurrent RL methods, especially in the context of recurrent off-policy RL. This issue mainly comes from the non-parallelizable nature of traditional RNNs. In our work, we tackle this problem by using the SSM structure, i.e. Mamba to improve the computational efficiency. Mamba is able to process sequential data in parallel based on the technique of parallel associative scan. In the following parts, we will analyze the time cost of Mamba compared with that of GRU.

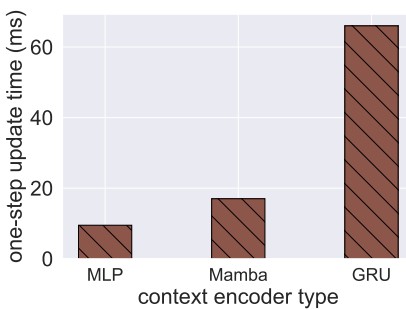

Figure 15: One-step update time with different types of context encoder.

To evaluate the improvement in training speed of the policy, we conducted experiments utilizing the HalfCheetah environment within Gym [53]. Maintaining consistency across all training configurations, we substituted Mamba in the context encoder with MLP and GRU, ensuring identical input/output dimensions for policy training. Each experiment contains 10,000 time steps, during which we measure the average training time per step. Policy training employed full-length trajectories, i.e. a fixed sequence length of 1000. The time costs associated with different network architectures are depicted in Fig. 15.

The results indicate that Mamba possesses significantly lower time costs compared to GRU, amounting to approximately 25% of the time taken by GRU. Mamba's time cost surpasses that of MLP by approximately 80%; nevertheless, it still presents a substantial speedup relative to GRU. These results are based on utilizing a single layer/block.

Table 3: GPU utilization, memory, and time cost with various neural network types. Time denotes the time cost for each update iteration, in HalfCheetah-v2. Normalized time is normalized with the corresponding GRU time cost.

| Network Type | GPU Utilization | GPU Memory (MB) | Time (ms) | Normalized Time |
|---|---|---|---|---|
| 1 Layer/Block | | | | |
| FC | 35% | 1474 | 9.5 | 14.4% |
| Mamba | 33% | 1812 | 17 | 25.8% |
| GRU | 58% | 1500 | 66 | 100.0% |
| 2 Layers/Blocks | | | | |
| FC | 37% | 1496 | 9.8 | 9.3% |
| Mamba | 38% | 1840 | 22 | 21.0% |
| GRU | 75% | 1558 | 105 | 100.0% |
| 3 Layers/Blocks | | | | |
| FC | 37% | 1500 | 10.5 | 7.1% |
| Mamba | 40% | 1912 | 29 | 19.7% |
| GRU | 77% | 1592 | 147 | 100% |
| 4 Layers/Blocks | | | | |
| FC | 38% | 1502 | 11 | 5.7% |
| Mamba | 43% | 1982 | 34 | 17.7% |
| GRU | 78% | 1612 | 192 | 100.0% |

**Time Overhead Comparisons with Different Number of Layers/Blocks.** In Table 3, the GPU utilization and time costs for different numbers of layers are detailed, complementing Fig. 15. The results indicate that the normalized time for Mamba decreases as the layer count increases, suggesting that Mamba becomes more computationally efficient with more blocks, compared with GRU. Additionally, Mamba consistently uses less GPU resources than GRU, further demonstrating its superior scalability and efficiency.

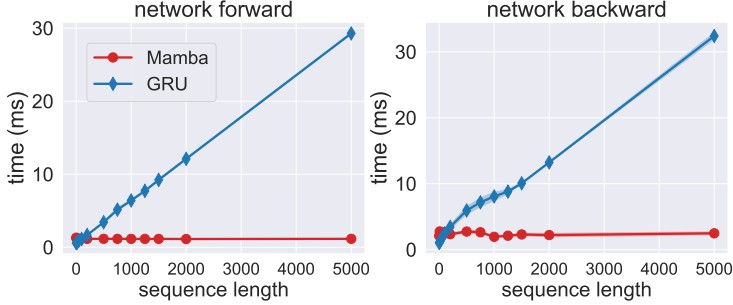

Figure 16: Network forward/backward time overhead for GRU and Mamba layers.

**Time Overhead Comparisons with Different Sequence Lengths.** To evaluate how Mamba scales with varying sequence lengths, we compared the forward and backward time costs of Mamba and GRU using artificial data. The results, presented in Fig. 16, show the sequence length on the horizontal axis. The left panel illustrates forward inference time, while the right panel depicts backward inference time. The time cost for GRU increases linearly with sequence length in both forward and backward scenarios. In contrast, Mamba's time cost remains relatively constant, showing no significant growth in either scenario. These results highlight Mamba's computational efficiency and scalability with respect to sequence length.

### E.3    More Comparative Results

#### E.3.1    POMDP Tasks

Table 4: Average performance on the classic POMDP benchmark with gravity changes at 1.5M time steps $\pm$ one standard error over 6 seeds.

|  | RESeL (ours) | PPO-GRU | MF-RNN | SAC-Transformer | SAC-MLP | TD3-MLP | GPIDE-ESS | VRM | A2C-GRU |
|---|---|---|---|---|---|---|---|---|---|
| AntBLT-P-v0 | **2829 ± 56** | 2103 ± 80 | 352 ± 88 | 894 ± 36 | 1147 ± 49 | 897 ± 83 | 2597 ± 76 | 323 ± 37 | 916 ± 60 |
| AntBLT-V-v0 | **1971 ± 60** | 690 ± 158 | 1137 ± 178 | 692 ± 89 | 651 ± 65 | 476 ± 114 | 1017 ± 80 | 291 ± 23 | 264 ± 60 |
| HalfCheetahBLT-P-v0 | **2900 ± 179** | 1460 ± 143 | 2802 ± 88 | 1400 ± 655 | 970 ± 47 | 906 ± 19 | 2466 ± 129 | −1317 ± 217 | 353 ± 74 |
| HalfCheetahBLT-V-v0 | **2678 ± 176** | 1072 ± 195 | 2073 ± 69 | −449 ± 723 | 513 ± 77 | 177 ± 115 | 1886 ± 165 | −1443 ± 220 | −412 ± 191 |
| HopperBLT-P-v0 | **2769 ± 85** | 1592 ± 60 | 2234 ± 102 | 1763 ± 498 | 310 ± 35 | 490 ± 140 | 2373 ± 568 | 557 ± 85 | 467 ± 78 |
| HopperBLT-V-v0 | 2480 ± 91 | 438 ± 126 | 1003 ± 426 | 240 ± 192 | 243 ± 4 | 223 ± 28 | **2537 ± 167** | 476 ± 28 | 301 ± 155 |
| WalkerBLT-P-v0 | **2244 ± 93** | 651 ± 156 | 940 ± 272 | 1150 ± 320 | 483 ± 86 | 505 ± 32 | 1502 ± 521 | 372 ± 96 | 200 ± 104 |
| WalkerBLT-V-v0 | **1901 ± 39** | 423 ± 89 | 160 ± 38 | 39 ± 18 | 214 ± 17 | 214 ± 22 | 1701 ± 160 | 216 ± 71 | 26 ± 5 |
| Average | **2471** | 1053 | 1462 | 716 | 566 | 486 | 2010 | −190 | 264 |

In Table 4, the corresponding final returns in Fig. 7 are listed in detail. We can find that RESeL makes notably enhancement and surpasses the previous SOTA (GPIDE) by $22.9\%$ in terms of the average performance.

#### E.3.2    Dynamics-Randomized Tasks

Table 5: Average performance on the MuJoCo benchmark with gravity changes at 2M time steps $\pm$ one standard error over 6 seeds.

|  | RESeL (ours) | SAC-MLP | SAC-GRU | ESCP | PEARL | EPI | OSI | ProMP |
|---|---|---|---|---|---|---|---|---|
| Ant-gravity-v2 | **5949 ± 314** | 1840 ± 358 | 758 ± 98 | 3403 ± 478 | 4065 ± 293 | 854 ± 2 | 878 ± 62 | 1243 ± 95 |
| HalfCheetah-gravity-v2 | **8705 ± 733** | 6225 ± 1273 | 7022 ± 451 | 7805 ± 91 | 5453 ± 19 | 6053 ± 582 | 6874 ± 147 | 885 ± 104 |
| Hopper-gravity-v2 | **2846 ± 191** | 1237 ± 192 | 1669 ± 98 | 2683 ± 369 | 2171 ± 337 | 2193 ± 68 | 1756 ± 31 | 252 ± 11 |
| Humanoid-gravity-v2 | **6360 ± 621** | 3491 ± 569 | 3064 ± 528 | 3857 ± 8 | 3849 ± 152 | 2673 ± 196 | 3518 ± 664 | 423 ± 8 |
| Walker2d-gravity-v2 | **5866 ± 266** | 3242 ± 3 | 2098 ± 270 | 2983 ± 73 | 4284 ± 174 | 1901 ± 242 | 1467 ± 224 | 271 ± 2 |
| Average | **5945** | 3207 | 2922 | 4146 | 3964 | 2735 | 2899 | 615 |

In Table 5, the corresponding final returns in Fig. 8 are listed in detail. We can find that RESeL makes notably enhancement and surpasses the previous SOTA (ESCP) by $43.4\%$ in terms of the average performance.

Table 6: Average performance on the classic MuJoCo benchmark at 300k, 1M, and 5M time steps, over 6 trials ± standard errors.

|  | Time step | TD3 | SAC | TQC | TD3+OFE | TD7 | RESeL (ours) |
|---|---|---|---|---|---|---|---|
| HalfCheetah-v2 | 300k | $7715 \pm 633$ | $8052 \pm 515$ | $7006 \pm 891$ | $11294 \pm 247$ | $\mathbf{15031 \pm 401}$ | $9456 \pm 232$ |
|  | 1M | $10574 \pm 897$ | $10484 \pm 659$ | $12349 \pm 878$ | $13758 \pm 544$ | $\mathbf{17434 \pm 155}$ | $12327 \pm 500$ |
|  | 5M | $14337 \pm 1491$ | $15526 \pm 697$ | $17459 \pm 258$ | $16596 \pm 164$ | $\mathbf{18165 \pm 255}$ | $16750 \pm 432$ |
| Hopper-v2 | 300k | $1289 \pm 768$ | $2370 \pm 626$ | $3251 \pm 461$ | $1581 \pm 682$ | $2948 \pm 464$ | $\mathbf{3480 \pm 22}$ |
|  | 1M | $3226 \pm 315$ | $2785 \pm 634$ | $\mathbf{3526 \pm 244}$ | $3121 \pm 506$ | $3512 \pm 315$ | $3508 \pm 522$ |
|  | 5M | $3682 \pm 83$ | $3167 \pm 485$ | $3462 \pm 818$ | $3423 \pm 584$ | $4075 \pm 225$ | $\mathbf{4408 \pm 5}$ |
| Walker2d-v2 | 300k | $1101 \pm 386$ | $1989 \pm 500$ | $2812 \pm 838$ | $4018 \pm 570$ | $\mathbf{5379 \pm 328}$ | $4373 \pm 144$ |
|  | 1M | $3946 \pm 292$ | $4314 \pm 256$ | $5321 \pm 322$ | $5195 \pm 512$ | $\mathbf{6097 \pm 570}$ | $5410 \pm 176$ |
|  | 5M | $5078 \pm 343$ | $5681 \pm 329$ | $6137 \pm 1194$ | $6379 \pm 332$ | $7397 \pm 454$ | $\mathbf{8004 \pm 150}$ |
| Ant-v2 | 300k | $1704 \pm 655$ | $1478 \pm 354$ | $1830 \pm 572$ | $\mathbf{6348 \pm 441}$ | $6171 \pm 831$ | $4181 \pm 438$ |
|  | 1M | $3942 \pm 1030$ | $3681 \pm 506$ | $3582 \pm 1093$ | $7398 \pm 118$ | $\mathbf{8509 \pm 422}$ | $6295 \pm 102$ |
|  | 5M | $5589 \pm 758$ | $4615 \pm 2022$ | $6329 \pm 1510$ | $8547 \pm 84$ | $\mathbf{10133 \pm 966}$ | $8006 \pm 63$ |
| Humanoid-v2 | 300k | $1344 \pm 365$ | $1997 \pm 483$ | $3117 \pm 910$ | $3181 \pm 771$ | $\mathbf{5332 \pm 714}$ | $4578 \pm 509$ |
|  | 1M | $5165 \pm 145$ | $4909 \pm 364$ | $6029 \pm 531$ | $6032 \pm 334$ | $\mathbf{7429 \pm 153}$ | $7280 \pm 168$ |
|  | 5M | $5433 \pm 245$ | $6555 \pm 279$ | $8361 \pm 1364$ | $8951 \pm 246$ | $10281 \pm 588$ | $\mathbf{10490 \pm 381}$ |

### E.3.3 Classic MDP tasks

Table 6 provides comprehensive training results for MuJoCo MDP environments, extending the data from Table 1. Specifically, the table includes results recorded at 300k, 1M, and 5M time steps. Baseline results are primarily sourced from [43]. In environments such as Hopper, Walker2d, and Humanoid, RESeL achieves or surpasses the previous SOTA performance. Notably, RESeL may underperform compared to previous baselines at the 300k time step. However, its asymptotic performance eventually matches or exceeds the SOTA. This suggests that incorporating RNN structures might slightly reduce sample efficiency initially, as the observation space expands to include full historical data, requiring more optimization to find optimal policies. Nevertheless, RESeL ultimately leverages the enlarged observation space to achieve superior asymptotic performance.

### E.4 Ablation Studies on Context Length

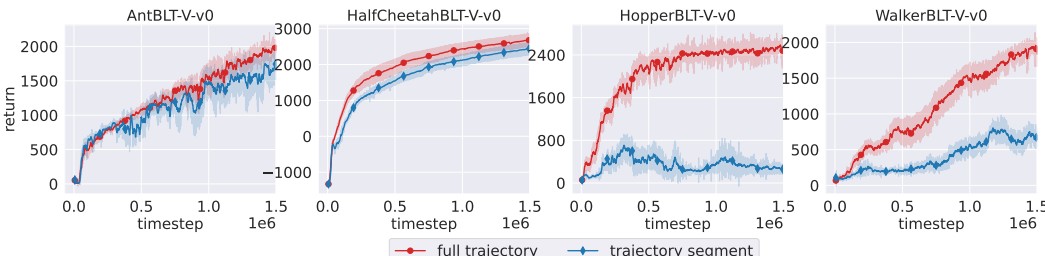

Figure 17: Ablation studies on the context length.

Since trajectory length (or called context length) in training data is often a critical parameter in previous studies [11], our work has emphasized the importance of using complete trajectories. Training with trajectory segments but deploying with full-trajectory length can lead to distribution shift issues. To investigate this, we conducted experiments using full trajectory data versus trajectory segments (64 steps). The results are illustrated in Fig. 17.

In general, models trained with full trajectory data perform better than those trained with trajectory segments. However, in the AntBLT-V and HalfCheetah-V environments, the performance gap is smaller. The key difference in these tasks is the longer trajectory lengths, especially in HalfCheetahBLT-V, where the trajectory length is fixed at 1000 steps. The data in these tasks are usually cyclic, while in other tasks, agents may drop and terminate trajectories before achieving stable performance. We suspect that strong cyclic data can mitigate distribution shift issues, as trajectory segments containing a complete cycle can effectively represent the properties of the full trajectory.

## E.5  More Comparisons of Policy Performance with different RNNs

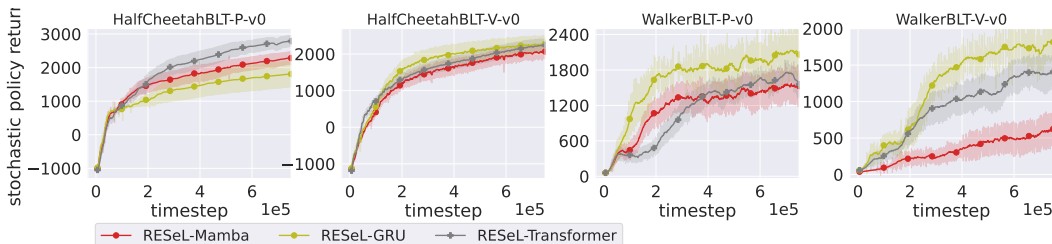

Figure 18: Learning curves in terms of the stochastic policy performance with different RNN architectures.

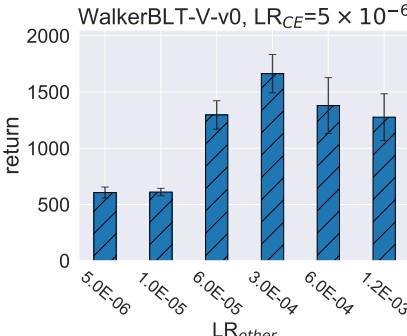

Figure 19: Sensitivity studies of the context-encoder-specific learning rates in terms of the average final return.

The learning curves of the exploration policy with difference RNN architectures are depicted in Fig. 18. Compared with Fig. 12, Fig. 18 includes evaluation with exploration noise added to the actions. In Fig. 18, RESeL-GRU shows a more substantial improvement over RESeL-Mamba than that in Fig. 12, suggesting that RESeL-GRU might perform better in tasks involving action disturbances.

## E.6  More Sensitivity Studies

Figure 19 serves as a supplementary panel to Fig. 11. In Fig. 19, we fixed $\mathrm{LR_{CE}} = 5 \times 10^{-6}$ and varied $\mathrm{LR_{other}}$ from $5 \times 10^{-6}$ to $1.2 \times 10^{-3}$. The results suggest that the optimal $\mathrm{LR_{other}}$ is still $0.0003$, which is significantly larger than $\mathrm{LR_{CE}}$. This finding further supports that the optimal learning rates for $\mathrm{LR_{CE}}$ and $\mathrm{LR_{other}}$ should be different and not of the same magnitude.

The learning rate sensitivity analysis on the remaining seven POMDP tasks is presented in Fig. 20. As shown, the conclusions in Sec. 5.3 still hold: when different learning rates are used for the MLP and RNN components and the MLP learning rate is fixed, excessively low or high RNN learning rates result in inefficient training. However, the optimal RNN learning rate generally falls between $5.0 \times 10^{-6}$ and $1.0 \times 10^{-5}$. If the MLP and RNN share the same learning rate, their performance rarely surpasses that of the setting with different learning rates, with the only exception being the HalfCheetahBLT-P-v0 environment, where a single learning rate slightly outperforms the different-learning-rate setting.

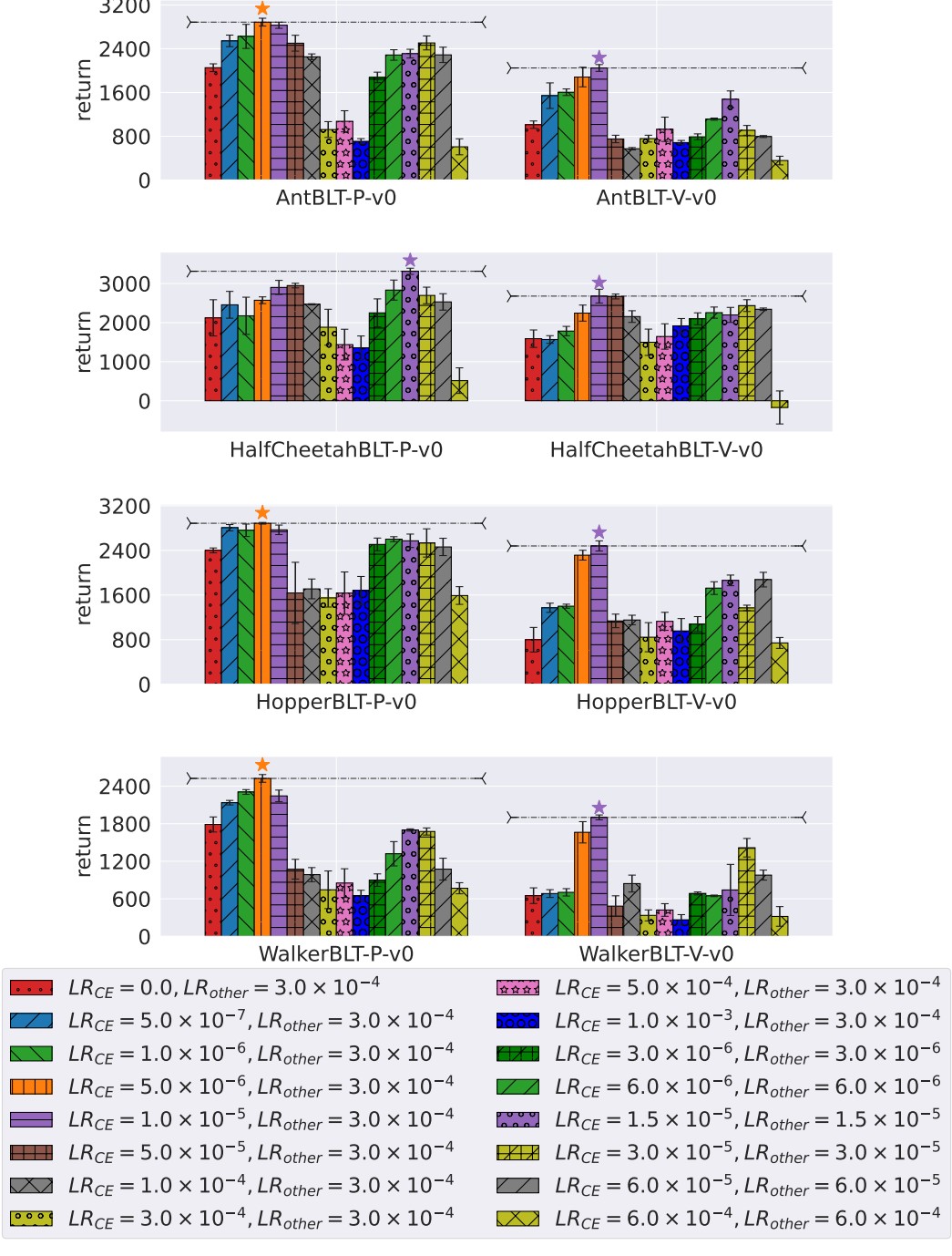

Figure 20: Sensitivity studies of the context-encoder-specific learning rates in terms of the average final return in eight POMDP tasks. The variants with the highest final return are marked with ★.

