# OpenReview forum: "Efficient Recurrent Off-Policy RL Requires a Context-Encoder-Specific Learning Rate"
_NeurIPS.cc/2024/Conference — NeurIPS 2024 poster_

### Official Review · Reviewer_B2hv · 2024-07-01

**Soundness:** 2
**Presentation:** 4
**Contribution:** 1
**Rating:** 3
**Confidence:** 4

**Summary:**

The authors investigate the stability of training deep recurrent policies for POMDP tasks. They hypothesize that the recurrent part of the encoder faces less stable training compared to the fixed-length parts of the encoder (e.g. the MLP input layer) and propose that the former should use a lower learning rate. This is shown to be effective mostly through an empirical evaluation.

**Strengths:**

The paper's ideas are communicated clearly and the authors present excellently summarize the state of the art for several partially observable settings, including the general setting, meta RL, and credit assignment. I appreciated the wide range of experimental domains and the sensitivity analysis demonstrating that the recurrent layers and the rest of the network should have different learning rates for optimal performance. The suggestion to introduce a new hyperparameter for the recurrent encoder could be broadly adopted.

**Weaknesses:**

Unfortunately, I believe the work is not sufficiently novel for acceptance at NeurIPS. The problem identified by the authors --- that a recurrent encoder is numerically less stable than a fixed-length encoder --- has been identified many times over the years with various techniques proposed to mitigate it. This includes classical works such as the original LSTM paper [1] aimed at tackling exponentially scaling gradients. More recently, techniques such as gradient clipping and truncating the number of recurrent backpropagation steps (which is not the same as shortening the context length) have been suggested to tackle this problem as well.

Experimentally, I would have liked to see that these last two tricks do not already solve the stability issues, as there was no mention of them in the paper. It also seems that the hyperparameters were tuned by hand (since I could not find any mention of a hyperparameter selection scheme) which I think is insufficient for a work for which the sole contribution is an empirical result that we are meant broadly apply. I would have expected to see a very rigorous and objective hyperparameter selection method such as a grid search over reasonable hyperparameters. While I appreciated the sensitivity analysis in Fig 9, it was only conducted on a single domain.

Ultimately however, my opinion is that the idea is not substantial enough for publication in a top venue.

[1]: Hochreiter, Sepp, and Jürgen Schmidhuber. "Long short-term memory." Neural computation 9.8 (1997): 1735-1780.

**Questions:**

1. How were hyperparameters tuned in the experiments?
2. Have you considered the approaches mentioned in the weaknesses? (gradient clipping and truncated gradients)

**Limitations:**

Yes.

---

> ### Author Rebuttal · Authors · 2024-08-07
>
> Thank you for taking the time to review our paper and providing us with your valuable feedback.
>
> - **[LSTM]** Our paper discovered a totally different issue from gradient exploding (numerical instability) resolved in LSTM: even if the RNN gradients do not explode and are similar in magnitude to those of MLPs, the gradient updates can still cause significant variations in the RNN outputs. These large output variations lead to instability in RL training, requiring a slow down in RNN updates. This is evident in Figure 3, particularly when the rollout step is 0. The action variation of the brown curve is lower than that of the orange curve, indicating that the RNN parameter changes are small and its gradient is not large. However, as the rollout steps increase, the brown curve grows by several orders of magnitude, which can destabilize reinforcement learning training.
>
> - **[The gradient clipping & truncated gradient tricks]**
>   - We have tried gradient clipping (0.5 maximum gradient norm as did in Dreamer) and truncated gradient (gradient maximum horizon 32) tricks, but we found that these tricks are irrelevant  to the problem of RNNs in RL. Their purpose is to prevent gradient explosion in RNNs and cannot slow down the RNN updates. In Fig. R1, we show the effects of these tricks. It can be seen that gradient truncation may still lead to instability and infinite values in tasks like Ant and HalfCheetah. Gradient clipping can suppress extreme values but does not improve performance. We conducted an in-depth investigation and compared the policy gradient norms of RESeL and the gradient clipping approach. As shown in Fig. R3, the policy gradient norm of the gradient clipping method has many spikes, reflecting training instability. This instability does not stem from gradient value instability, as we have already employed gradient clipping.
>
>   - We find the gradient instability originates from the instability of value function learning. In Fig. R4, we observe that in the baseline with gradient clipping but unchanged learning rates, the value loss frequently encounters outliers. This is due to large variation in the outputs of the policy and value networks, which make value function learning, based on bootstrapping, more unstable. This instability results in large value loss, which in turn causes spikes in the policy gradients. This issue is distinct from the instability and explosion of RNN gradients.
>
> - **[Sensitivity analysis was only conducted on a single domain]**  We have conducted similar experiments in WalkerBLT-P-v0 and AntBLT-V-v0. The results are shown in Fig. R2. Limited by computational resources, we only included the setting of Figs. 9(a) and 9(c) to illustrate that the optimal $LR_{CE}$ is around $10^{-5}$ in these tasks, and to show that $LR_{CE} = LR_{MLP}$ does not achieve optimal performance. The conclusions drawn from Fig. R2 are consistent with those presented in Figs. 9(a) and 9(c).
> - **[Hyper-parameters]** Some hyper-parameters (discount factor, reward as input) were chosen based on the nature of the environments, while others (Context encoder LR, policy and value function LR, and batch size) are determined via grid search:
>   - **Context Encoder LR**: We fixed the learning rate of the context encoder to be 1/30 of the policy learning rate in all tasks. We aim to have the action variation caused by RNN updates to be comparable to that caused by MLP updates, thus fixing the learning rate ratio between the context encoder and MLP. After conducting parameter searches in several environments, as shown in Figure 9, we find the 1/30 ratio performed well (this conclusion is also evident in Figure R2), so we fixed this ratio across all environments.
>   - **Policy and Value Function LR**: We mainly followed the default learning rate settings in CleanRL [1], using a policy learning rate of $3 \times 10^{-4}$ and an action-value function learning rate of $10^{-3}$. We conducted a grid search between the aforementioned learning rates and those reduced by a factor of $5$  (aiming to further stabilize training). We found that a smaller learning rate was more stable in tasks where the partial observability is not very significant (e.g., MDPs like classic MuJoCo).
>   - **Discount Factor ($\gamma$)**: For general tasks, we used a common $\gamma$ value of 0.99. For environments requiring long-term memory (e.g., Key-to-Door), we used a larger $\gamma$ of 0.9999.
>   - **Reward as Input**: The Classic Meta-RL tasks require inferring the true reward function based on real-time rewards. Thus, we also included real-time reward feedback as an input to the policy and value in these tasks.
>   - **Batch Size**: We conducted a grid search for batch size, mainly exploring sizes that were 1x or 2x the maximum trajectory length. We found that in classic POMDP tasks, a larger batch size performed better. In other tasks, the advantage of a large batch size was not significant and could even reduce training speed.
>
> [1] Huang et al. "CleanRL: High-quality single-file implementations of deep reinforcement learning algorithms." JMLR 2022.
>
> ----
>
> We sincerely appreciate your time and expertise in reviewing our paper. We would greatly appreciate it if you could re-evaluate our paper based on the above responses.

---

> ### Comment · Reviewer_B2hv · 2024-08-12
> **Re: Rebuttal**
>
> Thank you to the authors for the effort put into the rebuttal. I appreciate the inclusion of an investigation on gradient clipping and truncating the number of gradient steps, although I would have liked to see truncations with a smaller number of steps as well. Many implementations consider 4 or 8 steps (e.g. [1,2]) which may do well given that the output variation is exponential in the number of steps -- truncating to 32 steps doesn't provide much insight into whether it can tackle the exponential variation.
>
> I'm still not convinced about the novelty of the work after reading the authors' rebuttal and the other reviews. The authors claim that the novelty is not so much in their implementation trick but in the theoretical and intuitive understanding gained from this work. However, the idea that RNNs tend to be more unstable than MLPs is not a particularly new or surprising finding. Or is the novel understanding something different from what I just described?
>
> In reply to Reviewer 89BQ's argument for the novelty of this work, I can agree that the trick of using separate learning rates is not considered common practice. With a more rigorous investigation to ensure that current methods don't already solve this, such as gradient truncation with a smaller number of steps, I can agree that this knowledge could be useful to many practitioners (though perhaps it doesn't require a 9-page NeurIPS paper to disseminate). However, this paper doesn't seem to offer much beyond that. Most papers that propose a new algorithm also come with new and surprising insights that future works can build off of. Unfortunately, with this work, I don't think the insights are sufficiently novel. I don't see them pushing a research area forward, inspiring future works, or significantly improving our scientific understanding.
>
> I've increased by score since the experimentation was improved, but unfortunately my main complaint regarding lack of novelty still stands.
>
> [1] https://github.com/MarcoMeter/recurrent-ppo-truncated-bptt?tab=readme-ov-file
>
> [2] https://github.com/lcswillems/torch-ac?tab=readme-ov-file

---

> > ### Author Response · Authors · 2024-08-12
> >
> > Thank you for your reply. It appears that there may be a misunderstanding. The reviewer seems to believe that the instability in Recurrent off-policy RL is due to numerical explosion or instability of RNN gradients, and that addressing the gradient issues alone would resolve the overall training stability. This understanding aligns with the traditional issues of RNNs and corresponding solutions. However, there will be different issues in off-policy RL.
> >
> > Our experimental results (as shown by the red line in Fig. R3) indicate that the gradient explosion problem is not significant, as advanced RNNs (such as GRU, Mamba, etc.) have already addressed gradient instability quite effectively. If the gradients were indeed growing exponentially with the number of steps, merely reducing the RNN learning rate by 30 times would not be sufficient to counteract this exponential divergence, whereas gradient clipping would effectively mitigate such issues. Contrary to this, the experimental results show that our method is significantly more stable than gradient clipping.
> >
> > This is because the fundamental cause of instability in Recurrent off-policy RL is not gradient instability, but rather excessively large output variations between consecutive updates of the RNN. Even if the gradients are not large (on the same order of magnitude as those in MLPs), the outputs of the recurrent policy/value function can vary significantly between updates. These large output changes introduce instability into RL. For instance, in value function training, the (simplified) optimization target is $||Q(s,a)-r(s,a)-\gamma\hat{Q}(s',a')||_2^2$, where $\hat{Q}$ and $a'$ are the target value function network and the outputs of the current policy on $s'$, respectively. If the outputs of the value function network and policy network vary significantly, $a'$ will change greatly, and $\hat{Q}(s',a')$ will also fluctuate substantially. This results in large shifts in the optimization target of the value function after each update, leading to training instability. Similar instabilities occur in policy training, contributing to overall RL training instability. This is why, as shown in Fig. R4, even after clipping the gradients, the value loss remains unstable. **To the best of our knowledge, this finding has not been reported in previous work.** We will include these discussions in the revised version of our paper.
> >
> > Our theoretical results demonstrate that output variation does not increase exponentially with the number of steps but converges to a certain value. Therefore, we can balance the output variations, amplified by time steps, with a small RNN-specific learning rate. In contrast, if gradient clipping or truncation is used to keep the gradients within a normal range (similar to MLPs), the RNN output variations will still be large, and RL training will remain unstable.
> >
> > The results of Gradient truncation with 4 or 8 truncation steps are shown in the table below. In both variants, the algorithm produced infinite outputs before 250 iterations, leading to early stopping.
> >
> >
> >
> > |                     |           $LR_{CE}$=$10^{-5}$           | $LR_{CE}$=$3\times10^{-4}$, grad-step-truncation-4 | $LR_{CE}$=$3\times10^{-4}$, grad-step-truncation-8 | $LR_{CE}$=$3\times10^{-4}$, grad-step-truncation-32 |
> > | :-----------------: | :-------------------------------------: | :------------------------------------------------: | :------------------------------------------------: | :-------------------------------------------------: |
> > |     AntBLT-V-v0     | $ \mathbf{1986} \pm\mathbf{73} ^\star$  |                    $408\pm 74$                     |                    $269\pm 88$                     |                      $499±55$                       |
> > | HalfCheetahBLT-V-v0 | $ \mathbf{2679} \pm\mathbf{176} ^\star$ |                   $-107\pm 320$                    |                   $-703\pm 330$                    |                     $−458±244$                      |

---

> > > ### Comment · Reviewer_B2hv · 2024-08-12
> > > **Re: Additional Comment**
> > >
> > > Thank you for clarifying the difference between gradient instability vs large output variations and also for gathering further results for truncated BPTT. This is much appreciated.
> > >
> > > I now realize that I misinterpreted the equation in Proposition 1 to mean the output variation exponentially increases with $t$ when in fact it is bounded by a constant factor. I will reassess the work with this in mind. It would help if you could answer the following questions:
> > >
> > > - Could you give insight on the scope of the results with respect to off-policy RL vs RL in general vs supervised learning? The theoretical foundations don't seem particularly tied to off-policy RL, and the other paradigms also use neural architectures with recurrent and MLP layers. Is there any reason this work specifically targets off-policy RL?
> > >
> > > > To the best of our knowledge, this finding has not been reported in previous work.
> > >
> > > - When you mentioned the above statement, which finding are you specifically referring to? Is it that "the fundamental cause of instability in Recurrent off-policy RL is not gradient instability, but rather excessively large output variations between consecutive updates of the RNN"?

---

> > > > ### Author Response · Authors · 2024-08-12
> > > >
> > > > Thank you very much for your prompt response. We will address each of your concerns in detail.
> > > >
> > > > 1. **[The scope of the results]** We observed that significant RNN output variations can introduce instability in value learning when using bootstrap-based methods. This is because the target for value learning, $r(s,a) + \gamma \hat{Q}(s',a')$, experiences substantial changes. Off-policy RL algorithms such as TD3, SAC, TD7, and DQN, which heavily rely on value estimation, are especially prone to this instability. Additionally, in actor-critic off-policy RL methods, the optimization objective of the policy, $\max_{\pi} Q(s,a)$, also experiences large fluctuations, leading to even greater instability. On the other hand, on-policy algorithms, especially PPO and TRPO, have trust region constraints that prevent significant changes in policy outputs, which inherently mitigates the impact of RNN output variations. However the value function in these methods can still exhibit instability. We believe that incorporating the RESeL framework into on-policy algorithms could further enhance training stability. In supervised learning, the impact of large RNN output variations is less significant. This is mainly due to the use of fixed labels, which contributes to more stable training. Additionally, certain loss functions in supervised learning, such as MSE loss in the case of continuous labels, naturally suppress the issue of large RNN output variations as the gradient decreases rapidly with the loss. We will include this discussion in the revised version of the paper.
> > > >
> > > > 2. **[This finding]** Yes, we are the first to identify this fundamental cause of instability in recurrent off-policy RL.
> > > >
> > > > Please feel free to reach out if you need further clarification on any point. We are more than happy to answer any additional questions you may have.

---

### Official Review · Reviewer_vPkL · 2024-07-07

**Soundness:** 3
**Presentation:** 4
**Contribution:** 4
**Rating:** 7
**Confidence:** 4

**Summary:**

This paper proposes RESeL which improves recurrent off-policy RL in POMDPs mainly by applying a lower learning rate to the context encoder. This is justified by their theoretical analysis that the recurrence amplifies the output difference in the long run. In practice, they also incorporate several techniques (Mamba architecture, critic ensembles, efficient data sampling, slow policy update). The main contribution is on strong empirical performance. Across several benchmarks (classic POMDP, meta-RL, credit assignment), RESeL attains SOTA performance.

**Strengths:**

The problem setting of partial observability that this paper tackles is crucial and challenging. As indicated by the experiments, the partial observability is severe in some tasks and may require full context lengths (around 1000).

The paper is well written and easy to follow.

The performance gain averaged on a wide range of POMDP tasks is large enough to make RESeL a very strong empirical work. RESeL uses Mamba, which also accelerates training time a lot.

The ablation experiments on section 5.1 and 5.3 clearly shows that a smaller learning rate helps stability (through reduced action variation).

**Weaknesses:**

No major weaknesses. Please see the questions below.

One missing point is about the theoretical understanding -- why a smaller learning rate on context encoder can stabilize training? Reducing the output variation between consecutive updates, in my opinion, is a starting point, but not enough to explain the stability as a whole. This is connected with two-timescale update of feature vs value learning, e.g., https://openreview.net/forum?id=rJleN20qK7 Perhaps this related work is worth a discussion.

**Questions:**

About architectural design. Is there a reason for using respective context encoders in actor-critic?

About proposition 1. The connection between a smaller learning rate and the bound (1) is not very clear. Is the logic that a smaller learning rate leads to a smaller \|\theta - \theta’\| and thus a smaller \epsilon, then a tighter upper bound in (1)?

Is this a typo in the unusually large batch size described in Table 2?

As the main finding is on the learning rates, could the sensitivity analysis (Figure 9) be applied to more tasks?

Indicated by Figure 10, is RESeL-GRU more sample-efficient than RESeL-Mamba? Is Mamba a crucial component of RESeL?

**Limitations:**

Yes, no major limitations.

---

> ### Author Rebuttal · Authors · 2024-08-07
>
> We appreciate your time to review and provide positive feedback for our work.
>
> - **[Theoretical understanding]** Thank you very much for pointing out this relevant work. It has been very enlightening for us. RESeL and TTN are closely related, with the MLP value and the small learning rate RNN encoder corresponding to the value function and representation network, respectively. The convergence guarantee of TTN also explains why a small output variation between consecutive updates can enhance training stability.
>
> - **[Architectural design]** As noted in [1], sharing a context encoder could lead to a large gradient norm, causing instability. Therefore, we chose to use separate context encoders.
>
> - **[The connection between a smaller learning rate and the bound (1)]** You are right, reducing the learning rate can lower $\epsilon$, thereby reducing the upper bound. We will include this discussion in the revised version of our paper.
>
> - **[Large batch size described in Table 2]** The batch size here refers to the number of transitions. For each update, we collect at least one complete trajectory to calculate the loss. In environments like HalfCheetah, one trajectory consists of 1000 steps. So, the 1000 and 2000 mentioned correspond to one or two complete trajectories in these environments. To maintain consistency in the number of valid transitions across different environments, especially those with varying trajectory lengths, we use the same batch size for them. For these environment, we will dynamically sample uncertain number of complete trajectories to ensure that the number of valid transitions is no less than this batch size.
>
> - **[Sensitivity analysis in more tasks]** Yes, we have conducted similar experiments in WalkerBLT-P-v0 and AntBLT-V-v0. The results are shown in Fig. R2. Limited by computational resources, we only included the setting of Figs. 9(a) and 9(c) to illustrate that the optimal $LR_{CE}$ is around $10^{-5}$ in these tasks, and to show that $LR_{CE} = LR_{MLP}$ does not achieve optimal performance.
>
> - **[Comparison between RESeL-Mamba and RESeL-GRU]** It's true that GRU is more sample-efficient than Mamba in some environments. Mamba loses some expressive power due to its parallelzability (the hidden state at time $t$ is not used as input to the nonlinear network at time $t+1$ but is accumulated through a linear transition matrix) and also is not crucial in RESeL. However, Mamba significantly reduces computation time during training and requires much less memory in environments with variable trajectory lengths.
>
> [1] Ni et al. "Recurrent Model-Free RL Can Be a Strong Baseline for Many POMDPs." ICML 2022.
>
> ---
>
> Thank you for taking the time to provide us with your feedback. We appreciate your valuable comments and suggestions, which will help us improve our work. We look forward to receiving your further feedback.

---

> ### Comment · Reviewer_vPkL · 2024-08-13
>
> Thank you for the rebuttal and the additional experiments, although what I expected was a thorough sensitive analysis across all the tasks (currently only 3 tasks are shown).
>
> Overall the paper and the rebuttal looks good to me. I keep my rating.

---

> > ### Author Response · Authors · 2024-08-13
> >
> > Thank you for your response. Due to time and computational resource limitation, we were only able to add two additional environments during the rebuttal phase. We will continue with sensitivity studies and include the results and discussion of the remaining tasks in the next version of our paper. We appreciate your constructive suggestions, which have helped improve our paper.

---

### Official Review · Reviewer_dCQa · 2024-07-12

**Soundness:** 3
**Presentation:** 3
**Contribution:** 2
**Rating:** 5
**Confidence:** 3

**Summary:**

The paper mitigates the training instability issue of the recurrent off-policy RL by using a smaller learning rate for the RNN block.

**Strengths:**

- The paper is well-written and easy to follow.
- The paper proposes a simple solution with analysis.
- The proposed solution is thoroughly verified in different experimental settings.

**Weaknesses:**

- My main concern is the novelty of the proposed solution. Although the paper gives a reason for using different learning rate for the RNN block and the MLPs, the proposed method is to choose a lower learning rate for the RNN block. In practice, it's somehow common to choose different learning rate for different components. For example, Dreamer [Hafner, Danijar, et al.], which also uses an RNN block in their model, chooses a smaller learning rate for the RNN-based world model. Also, when facing the training stability issue, tuning the learning rate is always on the checklist.

    However, I would consider accepting the paper if it proposes an approach to automatically decide the learning rate of the RNN block by leveraging the analysis in section 4.2.
- Some details need further explanation:
    - In equation (1), it could be better to distinguish two $\epsilon$.
    - Will the amplification factor $\beta = \frac{K_y}{1-K_h}$ always larger than 1 (L160)?
    - The average variation in network output $\frac{1}{t}\sum^{t-1}_{i=0} || y_i - y'_i||$ converges to $\beta \epsilon + \epsilon$, which is not involved in $t$, but when this indicates with **longer** sequence lengths, the average variations in the RNN output induced by gradient deselect are amplified (L163-164)?
    - In the caption of Fig 3, what does it mean by "after one-step gradient-update"? Does it mean one-gradient update per environment step?
    - In L218, why does the learning rate of MLP must be increased **twentyfold**?
    - In L221, "The right panel shows the green and blue curve remain at similar levels until the final time step". Where are the green and blue curves?

**Questions:**

See weakness above

**Limitations:**

The limitation is discussed.

---

> ### Author Rebuttal · Authors · 2024-08-07
>
> Thank you for taking the time to review our paper, and for your insightful comments.
>
> - **[Dreamer]** It is not a common practice in reinforcement learning to use different learning rates for specific layers in a neural network. Typically, in reinforcement learning, the number of learning rates corresponds to the number of loss functions, with each loss function's associated network parameters sharing the same learning rate. For instance, in Dreamer, the world model is updated based on a prediction loss. Despite comprising various modules such as the non-RNN encoder, decoder, predictor, and RNN sequence model, these modules share a single learning rate in Dreamer families [1-3]. However, Figure 9(c) demonstrates that using the same learning rate for all layers during RL policy training could be suboptimal. Conversely, RESeL assigns a separate smaller learning rate specifically for the sequence model, while using a different learning rate for others.
>
> - **[Tuning the learning rate is always on the checklist]** We are not naively tuning the learning rate here. We have shown that uniformly tuning the overall learning rate does not achieve optimal performance. Instead, we propose RESeL to specifically set a smaller learning rate for the RNN, based on our theoretical analysis and intuitive understanding. Despite being straightforward, it is not trivial.
>
> - **[An approach to automatically decide the learning rate of the RNN block]**
>
>   - In all experiments, the learning rate for the RNN was set to 1/30 of the policy learning rate. We found this setting to be highly generalizable. From a practical standpoint, we recommend setting it to 1/30 of the policy learning rate or conducting a grid search centered around this ratio.
>
>   - On the other hand, for long trajectories, the upper bound of the average change in RNN outputs will eventually converge to $(1+\beta)\epsilon_{RNN}$, where $\epsilon_{RNN}$ is the variation of the RNN’s first step output. Our goal is to find a scaling factor $\alpha$ for the RNN learning rate so that the variation in RNN outputs matches $\epsilon_{MLP}$: $\alpha(1+\beta)\epsilon_{RNN}\approx\epsilon_{MLP}$. However, the values of $\beta$, $\epsilon_{RNN}$, and $\epsilon_{MLP}$ are highly dependent on the neural network weights. Thus, we can periodically test $(1+\beta)\epsilon_{RNN}$ and $\epsilon_{MLP}$ and use their ratio to determine $\alpha$. Specifically, for every 500 gradient updates, we perform a pseudo-update to observe how much the outputs of the policy change after individually updating the RNN block or the MLP. We then use the ratio of these changes to scale the RNN learning rate. To avoid instability in training caused by frequent changes in the learning rate, we only adjust the learning rate during the initial 50 iterations (a warmup phase, corresponding to 50000 gradient updates) and keep it fixed thereafter. We set the initial RNN learning rate to $3\times10^{-4}$. Our experimental results, shown in Fig. R6, indicate that the automated tuning method performs similarly to RESeL. We will add this discussion in our revised version. The final RNN learning rates are listed in the following table, which is close to our setting.
>
>     | Env. Name           | Auto-Tuned RNN Learning Rate            |
>     | ------------------- | --------------------------------------- |
>     | AntBLT-V-v0         | $6.0\times 10^{-6}\pm 3\times 10^{-7}$  |
>     | HalfCheetahBLT-V-v0 | $1.1 \times 10^{-5}\pm 1\times 10^{-7}$ |
>     | HopperBLT-V-v0      | $1.1 \times 10^{-5}\pm 4\times 10^{-7}$ |
>     | WalkerBLT-V-v0      | $1.2 \times 10^{-5}\pm 1\times 10^{-7}$ |
>
>
>
> - **[Distinguish two $\epsilon$]** Thank you for the suggestion. We will distinguish the two $\epsilon$ in the revised version.
>
> - **[Is $\beta$ always larger than 1]** It depends on the weights and gradient magnitude of the neural network. However, $\beta$ is always larger than $0$, so the upper bound of $||y_i-y_i'||(i>0)$ is always greater than $\epsilon$.
>
> - **[Average variation is not involved in $t$]** This needs to be understood in conjunction with Eq. (11) in Appendix B. Since the right-hand side of Eq. (11) is an increasing function of $t$, the mean value increases over time. We will include this explanation in the revised paper.
>
> - **[After one-step gradient-update in Fig. 3]** The action variation quantifies the difference in policy output after the gradient update compared to its output before the update, as considered in proposition 1. "After a one-step gradient update" indicates that we updated the policy only once, but we compare the differences in action outputs at various rollout steps before and after the policy update.
>
> - **[Learning rate of MLP must be increased twentyfold]** We found that increasing the MLP learning rate twentyfold to 0.006 resulted in action variation comparable to a CE learning rate of 0.0003. We will revise this statement in the updated paper to avoid confusion.
>
> - **[The green and blue curve]** This was a typo, it should actually refer to the orange and purple curves.
>
> [1] Hafner et al. "Dream to control: Learning behaviors by latent imagination." arXiv preprint 2019.
>
> [2] Hafner et al. "Mastering atari with discrete world models." arXiv preprint 2020.
>
> [3] Hafner et al. "Mastering diverse domains through world models." arXiv preprint 2023.
>
> ---
>
> We hope that the above response can address your concerns adequately. We would greatly appreciate it if you could re-evaluate our paper based on the above responses.

---

> > ### Comment · Reviewer_dCQa · 2024-08-12
> >
> > Thanks for the detailed reply and the efforts in preparing the rebuttal. After reading your rebuttal, I agree that although only a separate learning rate is used for RNN, the paper explains the motivation behind this and supports the conclusion with good experiments. I would improve my score to borderline accept.

---

> > > ### Author Response · Authors · 2024-08-13
> > > **Thank you**
> > >
> > > We are glad that our rebuttal was able to address your concerns. Your suggestions have been very helpful to us. On the other hand, should you have any additional questions or lingering concerns, please do not hesitate to contact us.

---

### Official Review · Reviewer_89BQ · 2024-07-25

**Soundness:** 3
**Presentation:** 3
**Contribution:** 3
**Rating:** 7
**Confidence:** 4

**Summary:**

The paper contributes to the important and long-standing issue of representing latent state information for successfully finding optimal RL control policies in the context of partially observable Markov decision processes (POMDP). The major contribution is a newly composed RL algorithm (RESeL), which, by design, enables a thorough study of the impact and importance of the actual learning rate (LR) used for the recurrent (latent) context encoder part of the solution architecture. The author highlight the fact that current state-of-the-art (SOTA) methods commonly use an equal LR for training both the latent representation and the policy/critic networks.

Their idea of separating the LR between recurrent CE and policy networks, as well as the overall performance benefits w.r.t current SOTA algorithms, is demonstrated by an extensive survey of selected POMDP tasks and benchmarks. These experiments together with theoretical arguments let the authors conclude that using a single  (and often too large) LR parameter approach of previous methods leads to sub-optimal performance results in solving POMDP tasks.

**Strengths:**

The paper presents a sound, well-written and focused approach of studying the impact of using dedicated LR parameters for both the context encoding and policy/critic related NN representation as part a newly proposed RL algorithm (RESeL) to solve POMDP tasks with high performance. It is implicitly stated (by not mentioning previous work) that the authors' work represents the first time that such a clear separation and impact analysis of different LR between the major architecture components is made, which - if true - points to a relatively high degree of originality.

The authors underline their central proposition (of better using lower LR for the CE related network parts) both by theoretical arguments/proof and an extensive experimental study on a relatively large set of well-known POMDP tasks and benchmarks, including direct comparisons to a large variety of SOTA algorithms. This presentation, together with additional material in the appendix, is well-suited to allow reproducibility and comparability of their results.

**Weaknesses:**

Even though mentioned in the Limitiation section, it would have been very helpful to add some benchmark comparisons of the RESeL algorithm with its RNN-CE block entirely bypassed/skipped to clearly separate the impact of the latent space presentation from the rest of the RL algorithm. In particular for the classic POMDP tasks, where only position and velocity features are masked, respectively, it is important to know a baseline performance when simply feeding the current and last-step observation vector into the MLP networks (given that RESeL uses 2 x 256 neurons layers in their MLPs, adding a few more inputs show be more than tractable). Including such extension (not necessarily for all benchmarks but maybe a smaller subset) could significantly improve the scientific insight.

Having clear focus on the impact of varying LR the remainder of the RESeL algorithm is entirely built from existing RL building blocks, like the overarching SAC RL architecture or convential RNN/GRU components and, hence, provides no original contribution in itself (as correctly mentioned by the authors, using RNN for latent space representation is known for a long time, as is the SAC approach). Re-using existing building blocks per-se is not a bad idea (and I don't recommend addressing this issue in the rebuttal phase) but one has to accept that the algorithmic novelty in this work is hence limited.

As a minor remark: the authors spend a relatively large portion of their introduction on the mathematical formulation of Proposition 1, whose central statement (variation of RNN outputs grow as function of rollout-step and LR) appears not entirely original (if it really is a first-time, the authors should highlight this fact more strongly) since the result is quite intuitive for people working in the RNN/RL community. Besides that, the connection between lowering the LR as part of the update rule when training RNNs and the resulting formula of Proposition 1 is not obvious and could be highlighted more strongly.

**Questions:**

l. 127: Why is the last-step observation also needed as input to the MLP if both current obs and context-encoded obs are already fed into the  NN? According to Fig. 2, the last-step obs is fed into the CE only (but not into the MLP)

l. 132: Is there any justification/reasoning behind the choice of the MLP architecture, in particular why a relatively large capacity (2 x 256 neurons) was chosen? Is there an estimate of how much this choice impacts the overall performance results of the policy training?

l. 163-164: It seems to me that the claim of Proposition 1 and the subsequential statements ("average variations in the RNN output ... are amplified") are not a novel discovery but have been studied and found by others in similar or different contexts of RNN before. In that case, it would be good to provide corresponding citations of pior art or - if the others believe this is the first time such a claim was made - highlight the significance of their finding.

l. 165-166: Please provide further arguments/proof, why the effect of Proposition 1 can be mitigated by smaller learning rate (even if such claim sounds plausible, it would be good to refer to previous work or more detailed reasoning). In other words: how does the learning rate affect the result of ||y - y'|| as a result of Propsotion 1 (it is likely related to epsilon in the formula but if so this relation could be mentioned more explicitly)?

l. 167-168: Similar remark as before: this sentence remarks a general claim about MLPs which should either be backed up by reference citations or by stronger reasoning or explanations.

l. 195: Given the typical level of stochasticity or dependency on random intial states, a trial number of 6 seems relatively low to incorporate statistical fluctuations in the return evaluation, in particular for the following ablation studies. Have the authors made sure that their reported results are not prone to larger statistical errors? What in detail is included in the choice of "different random seeds"? Does it only refer to the initial weight setup of the RNNs/MLPs or does it reflect varying intial conditions of the environemnts? Do all environments provide deterministic rollouts?

Figure 3: Why are error bars only visible for the brown curves/datapoints?

l. 218: How is "action variation after one-step update" as a function of rollout steps defined in the case of MLPs as they don't have an intrinsic time-convolution? I.e., what is the meaning of the i-th rollout time step in the case of MLP "only" (LR_CE = 0)? And why is that variation not close to zero for the 0-th rollout step, as indicated in Figure 3 (gray dashed curve) but starting from a value around 0.6?

l. 223-229: At this point of studying the performance in various benchmarks as a function of LR, and also for some of the subsequent ablation studies of Sec. 5.3, it would be great to show a benchmark reference where the CE module is skipped or bypassed altogether, feeding the current and last obserivation only into the MLP networks but not using a CE representation of the latent state at all (while there is one case where LR_CE = 0 in Fig. 9a, this is still different from skipping the entire module and following the suggestion above). This would generally help to underline the significance of the CE for the chosen set of benchmarks and tasks.

Fig. 6: It appears that SAC does hardly profit from using a recurrent CE (SAC-GRU) if compared to the MLP variant (SAC-MLP); is there any explanation why? It could stress the importance of showing the RESeL performance with and without CE, as mentioned in my comment above

Minor remarks:

l. 126: Better don't use " ' " behind name of algorithm ("RESeL's") but speak of RESeL policy as a joint expression

l. 171: Printing style (formatting) of scientific number notation "3e - 4" looks somewhat odd; check if correct Math formula style was chosen in the LaTex document or even better use decimal notation ("$3 10^-4$") or capital "E" instead of "e". This applies to other instances of number formatting of the main text as well.

l. 240: SOTA (state-of-the-art) should be introduced as an abbreviation

Fig. 5 (and others): Quality of plots could generally be improved by avoiding overlapping of images and axes captions in some cases

Fig. 6: Colors/symbols of "EPI" and "SAC-MLP" are hardly distinguishable

**Limitations:**

The authors have addressed the main limitation well. As they state, what is missing is a study on the impact of using the RNN as a context encoder in general, which refers to my request for providing a reference baseline where the CE is bypassed.

---

> ### Author Rebuttal · Authors · 2024-08-07
>
> We sincerely appreciate the time and effort you have invested in reviewing our paper, as well as your insightful and constructive feedback. Your comments have greatly assisted us in improving the quality of our work.
>
> - **[Skipping CE blocks]** Thank you for pointing out the missing baseline. We implemented  this baseline by setting the CE outputs to zero while keeping everything else unchanged to isolate the effect of the latent space. The results are shown as the NO_CE curve in Fig. R1. The results indicate that historical information is crucial in our algorithm. Without it, achieving high returns on these POMDP tasks is hard.
>
> - **[Algorithmic novelty]** Our work primarily focuses on the impact of learning rates in recurrent RL. We believe the importance and ubiquity of this issue merit a dedicated discussion.
>
> - **[Originality of Proposition 1]** We referenced the proof from model imitation learning [1]. We consider the RNN as the transition function of an MDP and then, based on the derivation in [1], combined with the characteristics of RNNs, we derived the compounding error of RNNs. To the best of our knowledge, we are the first to derive the compounding error bound in RNNs. Thank you for your suggestion, we will highlight this point and include the relevant citations in the revised paper.
>
> - **[Relationship between Proposition 1 and reducing learning rate]** Lowering the learning rate reduces the single-step network output variation $\epsilon$, thereby lowering the upper bound of Eq. (1).
>
> - **[Last-step observation]** Here we use an MLP-based pre-encoder, which is not the MLP policy, to map the last-step observation to a latent space. The MLP policy does not take the last-step observation as input.
>
> - **[MLP architecture]** We adopted the MLP network design from algorithms like SAC [2] and TD7 [3], which use a two-layer MLP for the policy network, with each layer consisting of 256 neurons. We previously tried smaller network structures, we find the impact was not significant.
>
> - **[Claims on MLP]** A smaller learning rate makes MLP training very slow, requiring many gradient updates to train the MLP adequately, leading to inefficiency. We will add this explanation to the revised paper.
>
> - **[Random seed]**
>
>   - We set a random seed at the start of each experiment to randomly initialize everything, including neural network weights and the environment. Every 1000 gradient updates, we perform a policy evaluation without resetting the environment's random state (forking the current random state). Using the current policy, we deterministically collect five trajectories for evaluation, where the policy outputs mean actions, but the initial state of the environment is re-sampled at the start of each trajectory. We repeated the experiment six times, each with a different random seed.
>   - To verify that our choice of seed count does not introduce larger statistical errors, we conducted an additional experiment with six more seeds on WalkerBLT-V-v0 and HopperBLT-V-v0. The results, shown in Fig. R5, demonstrate that the new curve (seed_7-12) closely matches the mean and shadow of the old curve (seed_1-6), indicating no significant statistical deviation. This shows that six seeds are sufficient for our experimental setup.
>
> - **[Figure 3]** Regarding Figure 3, we trained the policy using RESeL and collected trajectories with non-deterministic actions in the environment. We obtained the mean actions $\{a_0,a_1,a_2,...\}$ for each trajectory, where $1,2,3,...$ are rollout steps. We then performed a single gradient update on the policy and obtained the updated policy's mean action outputs at each state in the trajectory $\{a_0', a_1', a_2', \ldots\}$. Figure 3 demonstrates $\{||a_0-a_0'||_2,||a_1-a_1'||_2,||a_2-a_2'||_2,...\}$ for different learning rates, with each experiment repeated 20 times. Although each curve has certain variance, the high repetition count results in a small standard error. The brown curves have a larger variance, hence the noticeable shadow (standard error), while other curves have smaller variance, making the shadow less apparent but present. In the gray dashed curve, the excessively high learning rate caused unpredictable behaviors, leading to significant initial variations in the policy. The gray dashed curve starts high and then decreases, possibly due to the larger action amplitude in the latter half of the trajectory, reaching the boundary values. As a result, many actions reach the boundary values and get compressed after substantial policy updates, leading to less variation compared to the first half.
>
> - **[SAC-GRU and SAC-MLP]** We believe the too large output variations of RNNs cause the value and policy learning process being instable, therefore preventing the policy from efficiently utilizing the historical information.
>
> - **[Minors]** Thank you for pointing these out. We will correct all these issues in the revised paper.
>
>
>
> [1] Xu et al. "Error bounds of imitating policies and environments." NeurIPS 2021.
>
> [2] Haarnoja et al. "Soft actor-critic: Off-policy maximum entropy deep reinforcement learning with a stochastic actor." ICML 2018.
>
> [3] Fujimoto et al. "For sale: State-action representation learning for deep reinforcement learning." NeurIPS 2023.
>
> ---
>
> Thank you for guiding us towards making it a better work. We hope our responses have addressed your concerns effectively and enhanced the clarity of our main contributions. We look forward to your further comments.

---

> > ### Comment · Reviewer_89BQ · 2024-08-08
> >
> > I'd like to thank the authors for their detailed response to my remarks and concerns, and in particular for their additional study of bypassing the CE to provide another baseline for the performance evaluation. Together with other explanations and clarifications done in the revised version my remaining questions and suggestions also have been sufficiently addressed. Hence, I continue to vote for "Accept (7)".
> >
> > Towards the main concern regarding lack of novelty, as raised in my own review and reflected by some of the other reviewers' remarks: It is true that tuning the LR in any context of machine learning is on the checklist of every good practioner, so I was tempted to come to the same conclusion in the beginning that this work doesn't add enough novelty regarding this aspect. What changed my final perspective is related to the answer given by the authors in their rebuttal to some of the other reviews: In the particular context of multi-component POMDP-RL solution architectures it is not common practice to use separate LR for modules which otherwise contribute to the same overall loss function. And even if that idea was previously applied in this particular context, it would highly likely not entail a comprehensive study as provided in the present work.

---

> ### Author Response · Authors · 2024-08-08
>
> Thank you very much for your prompt response and kind comments. We are delighted that our reply has addressed your concerns. Your valuable suggestions have greatly improved our work.

---

### Author Rebuttal · Authors · 2024-08-07

We appreciate the time and effort all the reviewers have dedicated to our paper and their highly constructive comments. Here, we provide a general response to the common concerns raised.

- **[Novelty/Contribution]**
  - We would like to emphasize that our core contribution lies in presenting a universal principle rather than specifying the hyperparameters for each task. This principle is supported by theoretical foundations and intuitive understanding, making it more than just an empirical trick. Although it is simple, it is by no means trivial.
  - Specifically, our findings indicate that the instability in recurrent RL arises from the amplified action variation by rollout step, specifically the compounding error in RNN output. This results in larger overall output variation for the RNN compared to the MLP, even when their single-step changes are the same, leading to instability during training. This paper proposes to solely slow down updates of the RNN by giving it a lower learning rate, then $\epsilon$ in Eq. (1) is reduced, thereby decreasing the RNN's output variation and improving the stability. There have been various traditional approaches that can improve the training stability of RNNs in non-RL scenarios, such as reducing the overall learning rate, gradient clipping, and gradient truncation. However, these methods cannot solely slow down the update of RNN thus still sufferring training instability in reinforcement learning.

- **[Relationship between Proposition 1 and reducing learning rate]**
  Reducing the learning rate can decrease the single-step network output variation, denoted as $\epsilon$, thereby reducing the upper bound of Eq. (1).

- **[Figure 3]**
  Here we elaborate on how we obtained Figure 3: We trained a policy using RESeL and collected a batch of trajectories in the environment with stochastic exploration. For each trajectory, we calculated the mean action output by the RESeL policy $\{a_0, a_1, a_2, \ldots\}$, where $1, 2, 3, \ldots$ represent rollout steps. We then performed a single gradient update on the policy and obtained the updated policy's mean action outputs at each state in the trajectory $\{a_0', a_1', a_2', \ldots\}$. In Figure 3, we plotted the action variation $\{||a_0 - a_0'||_2, ||a_1 - a_1'||_2, ||a_2 - a_2'||_2, \ldots\}$ under different learning rate settings. **The action variation means the difference in policy output after the gradient update compared to its output before the update.**


- **[RNN learning rate setting]**
  - We fixed the learning rate of the RNN to be 1/30 of the policy learning rate for all tasks. Fig. 9 and Fig. R2 demonstrate that this ratio achieves the highest policy performance. Additionally, Fig. 3 also shows that this ratio effectively prevents the RNN output variation from significantly exceeding that of the MLP. Our experiments indicate that this ratio has good generalizability and works well in all tasks. Therefore, we recommend setting the RNN learning rate to 1/30 of the policy MLP learning rate in practice or conducting a grid search centered around this ratio.
  - Based on the suggestion from Reviewer dCQa, we have also implemented an automatic RNN Learning Rate Tuning method. At the beginning of training, we introduce a warmup phase where the RNN’s learning rate is automatically adjusted. The tuning objective is ensuring that the action variation caused by updating the RNN alone is roughly consistent with that of the MLP.

**We have uploaded the additional experiment results (Figs. R1-R6) in a separate PDF.**

---

### Decision · Program_Chairs · 2024-09-25

**Decision:**

Accept (poster)

**Comment:**

While Reviewer B2hv votes for rejecting the paper due to insufficient depth, the three remaining reviewers vote in favor of acceptance provided extensive changes are made.
Both views are well-founded and comprehensible.  In my view, these views are best summarized as “Borderline Accept” and as there are no major flaws in the paper, it depends on where the bar is set for NeurIPS 2024 whether this paper can be accepted.